# Applications of Artificial Neural Networks in Greenhouse Technology and Overview for Smart Agriculture Development

**Axel Escamilla-García [1], Genaro M. Soto-Zarazúa [1],*, Manuel Toledano-Ayala [2], Edgar Rivas-Araiza [2] and Abraham Gastélum-Barrios [1]**

[1] Facultad de Ingeniería Campus Amazcala, Universidad Autónoma de Querétaro, Carretera a Chichimequillas S/N Km 1, Amazcala, El Marqués, Querétaro 76265, Mexico; aescamilla514@alumnos.uaq.mx (A.E.-G.); abraham.gastelum@uaq.mx (A.G.-B.)

[2] Facultad de Ingeniería Centro Universitario, Universidad Autónoma de Querétaro, Cerro de las Campanas S/N, Centro Universitario, Santiago de Querétaro, Querétaro 76010, Mexico; toledano@uaq.mx (M.T.-A.); erivas@uaq.mx (E.R.-A.)

* Correspondence: soto_zarazua@yahoo.com.mx; Tel.: +1-52-442 332 9 7 13

**Abstract:** This article reviews the applications of artificial neural networks (ANNs) in greenhouse technology, and also presents how this type of model can be developed in the coming years by adapting to new technologies such as the internet of things (IoT) and machine learning (ML). Almost all the analyzed works use the feedforward architecture, while the recurrent and hybrid networks are little exploited in the various tasks of the greenhouses. Throughout the document, different network training techniques are presented, where the feasibility of using optimization models for the learning process is exposed. The advantages and disadvantages of neural networks (NNs) are observed in the different applications in greenhouses, from microclimate prediction, energy expenditure, to more specific tasks such as the control of carbon dioxide. The most important findings in this work can be used as guidelines for developers of smart protected agriculture technology, in which systems involve technologies 4.0.

**Keywords:** artificial neural network; greenhouse; deep learning; optimization algorithms; hybrid neural networks; microclimate

## 1. Introduction

Greenhouses are systems that protect crops from factors that can cause them damage. They consist of a closed structure with a cover of translucent material. The objective of these is to maintain an independent climate inside, improving the growth conditions for increasing quality and quantity of products. These systems can produce in a certain place without any restriction of agroclimatic conditions. However, they must be designed according to the environmental conditions of the place where they will be installed. Control of the microclimate is necessary for optimal development of the plant since it represents 90% of the yield of crop production, where the equipment, shape, and elements of the greenhouse will depend on how different the outdoor climate is from requirements of the plant [1–4].

When speaking of the greenhouse climate, reference is made to the environmental conditions that the plants require to be in good condition [5]. The greenhouse microclimate is complex, multiparametric, non-linear and depends on a set of external and internal factors. External factors include meteorological factors such as ambient temperature and humidity, the intensity of solar radiation, wind direction, and speed among others. Internal factors are crops, greenhouse dimensions, greenhouse components and elements such as heating, fogging and ventilation systems,

soil types, etc. [6]. There are two different approaches to describe the greenhouse climate: one is based on energy and mass flow equations that describe the process [7–13] and the other consists of the analysis of input and output data of the process using a system identification approach [2,14–18]. However, even with these approaches, it is difficult to fully account for all of these factors. In this sense, it is desirable to solve the microclimate problem based on modern methods of non-linear and adaptive systems [19].

It is important to describe the greenhouse climate to design a good control system since it is a way of manipulating the variables that affect its behavior [20]. The greenhouse climate control provides a favorable environment for cultivation and this achieves predetermined and optimal results. Nowadays, several control techniques and strategies, such as predictive control [16,21–23], adaptive control [24–27], non-linear feedback control [28,29], fuzzy logic (FL) control [30,31], robust control [15,32–34] and optimal control [35–37] have been proposed for the control of the greenhouse environment. However, for the environmental control of greenhouses, conventional proportional, integral and derivative controllers (PID) are mainly developed due to their flexibility, architecture and good performance [38].

Another topic of interest derived from the production of greenhouse crops is energetic consumption, in which solar energy is presented as a viable substitute for traditional sources (fuel and electricity). Solar energy is better than traditional sources because fuels are not renewable and represent high cost [39]. Traditional energy sources can be replaced with other sustainable energies, such as solar energy, wind energy [40], biomass [41–43], geothermal energy [44–46], cogeneration systems [47,48], among others. However, use of solar photovoltaic cells or solar thermal energy in greenhouses are more widely used and can commonly be combined with other sustainable energy systems [49]. Solar greenhouses provide a controlled system cultivation, the most focus is to reduce heating energy requirements, i.e., the heating requirement is largely derived from the sun [39]. Furthermore, solar energy represents a primary element in the heating of greenhouses and makes it possible to minimize production costs [50]. Several studies have been carried out in which energy savings are sought, where methods such as genetic algorithms (GA) have been applied to optimize energy collection [51], also physical models [52–54], as well as computational fluid dynamics (CFD) techniques to predict the microclimate of solar greenhouses [55–57].

Prediction methods can be divided into two groups: physical methods based on mathematical theory, which requires a large number of parameters to be determined, as well as the difficulty of measuring those parameters; and black box methods based on modern computational technology (particle swarm optimization algorithm, least squares support vector machine model), which do not always guarantee convergence to an optimal solution and easily undergo partial optimization [58]. On the other hand, instead of being programmed, neural networks (NNs) learn to recognize patterns. These systems are highly appropriate to reflect knowledge that cannot be programmed or justified, as well as to represent non-linear phenomena [59]. Figure 1 presents the interest topics in greenhouses and the classification of the models used.

Within the latter, as presented, several studies have been developed for different applications in greenhouse crop production. However, since greenhouses are non-linear, invariant over time and with a strong coupling [15], several investigations have opted to use artificial neural networks (ANNs) for the simulation, prediction, optimization, and control of these processes. Mathematical analysis methods have been developed for the optimization of the ANNs database. These models present a variables relationship, make the variables less trivial, and simplify the structure of the network. In this way improve greenhouse total yield [60].

This review explores different ANNs investigations and applications in greenhouse technology. Presents trends for future research in the development of this type of model will improve its application and integration with the 4.0 technologies that are currently applied in smart agriculture (SA) but are little used in greenhouse production such as the internet of things (IoT), machine learning (ML), image analysis, big data, among others. The structure of this document is: Section 2 gives an explanation of ANNs, different activation functions, types and different knowledge about ANNs. Section 3 presents the NNs application for the prediction of microclimates in greenhouses.

Section 4 shows neural network applications in greenhouse energy optimization. Section 5 indicates other studies and ANNs in greenhouse applications. Finally, Section 6 addresses the challenges in the development of NNs in greenhouse agriculture.

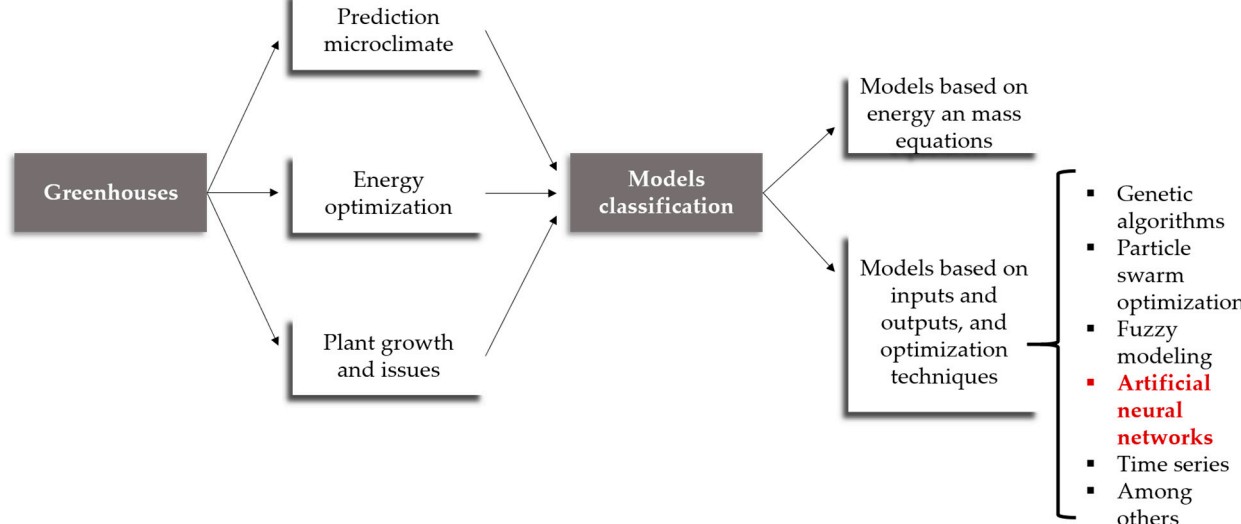

**Figure 1.** Interest topics in greenhouses and models classification. Genetic algorithms: GA; Particle swarm optimization: PSO; Artificial neural networks: ANNs

## 2. Artificial Neural Networks

An ANN is a ML algorithm based on the concept of a human neuron [13]. It is a biologically inspired computational model, consisting of processing elements (neurons) and connections between them with coefficients (weights) attached to the connections [61]. ANNs are inspired by the brain structure and for this reason it is important to define the main components under which a neuron, dendrites, cell body, and axon works. Dendrites are a network that carries electrical signals to the cell body. The cell body adds and collects the signals. The axon carries the signal from the cell body to other neurons using a long fiber. When the axon of a cell comes in to contact with a dendrite of another cell it is known as a synapse. Therefore, the functions of neuronal networks are established through the arrangement of neurons and individual synaptic forces [62]. Figure 2 presents a general schematic of a biological neuron with each element that makes it up.

Neural structures develop through learning; however, they constantly change, strengthening or weakening the synaptic junctions. Although ANNs are inspired by the brain, they are not that complex. However, the greatest similarities are primarily that both networks are interconnected and the functions of the networks are determined by the connections between neurons [63].

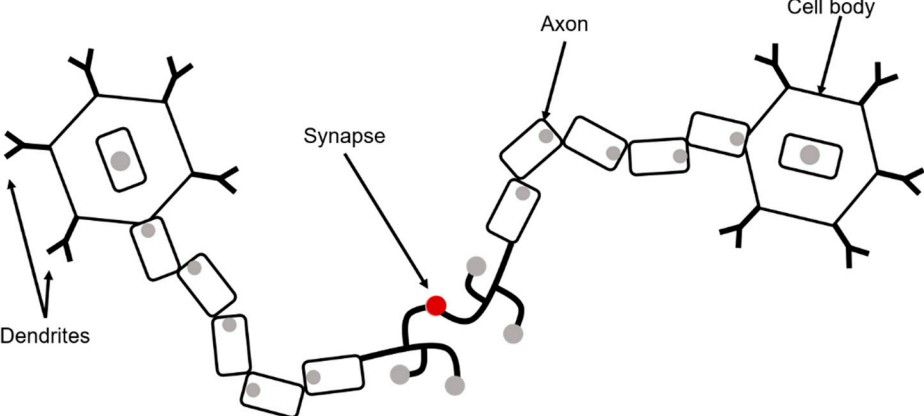

**Figure 2.** Structure of a biological neuron.

Neurons receive inputs such as impulses. The peak rate generated over time and the average peak generation rate in several runs, are some measures used to describe neuron activity. In ANNs, a neuron is identified by the speed at which it generates these peaks. A neuron connects to other neurons in the previous layer through adaptive synaptic weights. Knowledge is generally stored as a set of connection weights. When these connection weights are modified in an orderly manner and with a suitable learning method, a training process is carried out. The learning method consists of presenting the input to the network and the desired output, adjusting the weights so that the network can produce the desired output. After training the weights will have relevant information, whereas before training it is redundant and meaningless [64].

Figure 3 presents the simple neuron structure. The processing of the information in a neuron begins with the inputs $X_n$, they are weighted and added up before going through some activation function to generate its output, this process is represented as $\xi = \Sigma X_i \cdot W_i$. For each of the outgoing connections, this activation value is multiplied by the specific weight $W_n$ and transferred to the next node. If it considers a linear activation, the output would be given by *y=α(wx+b)* [65].

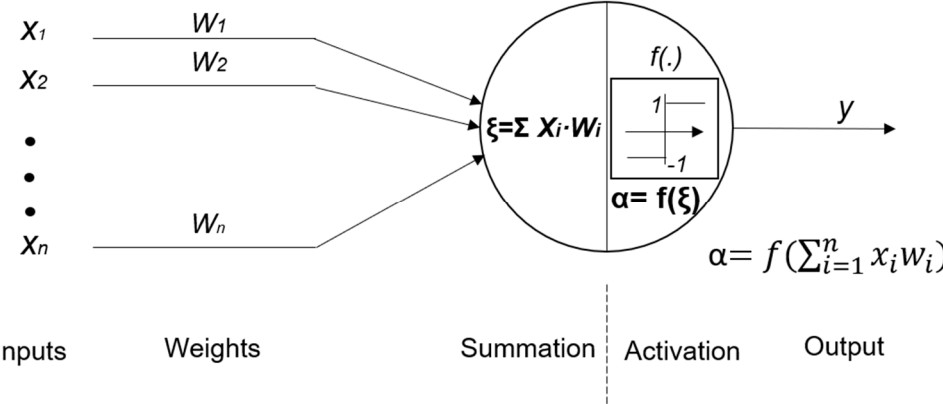

**Figure 3.** The basic scheme of a neuron.

### 2.1. The Activation Function of an Artificial Neural Network

The activation function is a function that receives an input signal and produces an output signal after the input exceeds a certain threshold. That is, neurons receive signals and generate other signals [66]. The neuron start is only performed when the sum of the total inputs is greater than the neuron threshold limit, then the output will be transmitted to another neuron or environment. This threshold limit determines whether the neuron is activated or not, the most common activation or transfer functions are the linear, binary step, piecewise linear, sigmoid, Gaussian and hyperbolic tangent functions [67].

Table 1 shows the activation functions commonly used in NNs . The behavior of neurons is defined by these functions. If it transfers a function that is linear and the network is multi-layered, it can be represented as a single-layer network, since it is product of weight matrices of each layer and will only produce positive numbers over the entire range of real numbers. On the other hand, non-linear transfer functions (sigmoid function) between layers allow multiple layers to provide new capabilities, adjusting the weights to obtain a minimum error in each set of connections between layers [68,69]. Linear functions are generally used in the input and output layers, while non-linear activation functions can be used for the hidden and output layers [70].

The most used non-linear activation functions are sigmoid and hyperbolic tangents. Hyperbolic and sigmoid tangents are mainly used because they are differentiable and make them compatible with the back-propagation algorithm. Both activation functions have an "*S*" curve, while their output range is different [71]. The sigmoid function is the most used activation function in ANNs. This function varies from 0 to +1, although the activation function sometimes seeks to oscillate between −1 and +1, in which case the activation function assumes an antisymmetric form with respect to the origin, defining it as the hyperbolic tangent function [72–74].

Direct implementation of sigmoid and hyperbolic tangent functions in hardware is impractical due to its exponential nature. There are several different approaches to the hardware approximation of activation functions, such as the piecewise linear approximation. Linear part approximations are slow, but they are the most common way of implementing activation functions [75]. In addition, this uses a series of linear segments to approximate the trigger function. The number and location of these segments are chosen so that errors and processing time are minimized [76].

The Gaussian activation function can be used when finer control over the activation range is needed [69]. Furthermore, it can uniformly perform continuous function approximations of various variables [77].

**Table 1.** Activation functions for layers in artificial neural networks.

| Name | Graphic | Function | |
|------|---------|----------|---|
| Linear |  | $f(\xi) = a \cdot \xi + b$ | |
| Binary step |  | if $\xi \geq 0$, <br> if $\xi < 0$, | then $f(\xi) = 1$, <br> then $f(\xi) = 0$, |
| Piecewise linear |  | if $\xi \geq \xi_{max}$, <br> if $\xi_{min} > \xi > \xi_{max}$, <br> if $\xi \leq \xi_{min}$, | then $f(\xi) = 1$, <br> then $f(\xi) = a \cdot \xi + b$, <br> then $f(\xi) = 0$, |
| Sigmoid |  | $f(\xi) = \dfrac{1}{1 + e^{-b \cdot \xi}}$, | interval (0,1) |
| Gaussian |  | $f(\xi) = e^{-\xi^2}$, | interval (0,1] |
| Hyperbolic tangent |  | $f(\xi) = \dfrac{2}{1 + e^{-2 \cdot \xi}} - 1$, | interval [-1,1] |

*2.2. Types of Artificial Neural Network*

The ANNs are classified according to different criteria, we can establish that there are two types [78]:

- Feedforward neural networks (FFNNs);
- Recurrent neural networks (in discrete time) or differential (in continuous time);

### 2.2.1. Feedforward Neural Networks

The neuron is the basic component of NNs. Neurons are connected to each other through synaptic weight [62]. Considering a neural network with three layers such as in Figure 4: an input layer, a hidden layer and an output layer the intermediate layer is considered self-organized Kohonen map, which consists of two layers of processing units (input and output), depending on the complexity of the network (there may be several hidden layers in each network) [79]. In FFNNs, information progresses, from the input nodes to the hidden nodes and from the hidden nodes to the output nodes. When an input pattern is fed into the network, the units in the output layer compete with each other, and the winning output unit is the one whose input connection weights are closest to the input pattern, the number of neurons in the input and output layers is the same as the number of inputs and outputs of the problem [80]. The learning method can be divided into two stages, the first stage is to determine the neuron of the hidden layer whose weight vector is the first input vector and the second refers to the training process. Initially, the Euclidean distance between the input and the weight vector of the first neuron will be calculated. If the distance is greater than a predetermined distance threshold value, a new hidden-layer neuron is created by assigning the input as the weight vector. Otherwise, the input pattern belongs to this neuron. During training, each pattern presented to the network selects the closest neuron on a Euclidean distance measure, modifying the winner's weight vector, and topological neighbors draws them in the direction of the input, the weights leaving the winning neuron and its neighbors are adjusted by the gradient descent method [81]. Forward NNs fall into two categories based on the number of layers, either single layer or multiple layers [82].

Back-propagation (BP) is a type of ANN training, used to implement supervised learning, tasks for which a representative number of sample inputs and correct outputs are known. BP is derived from the difference in desired and predicted, output; this is calculated and propagated backward [83]. First, network weights to a small random weight are initialized, the vector set of input data to the network are presented, the input propagated to generate the output, which is called the input advance phase, and the error comparing the estimated net output with the desired output calculated [84]. The weight will be corrected from the output to the input layer that is, in the backward direction in which the signals propagate when objects are introduced into the network. This is repeated until the error no longer improves [85].

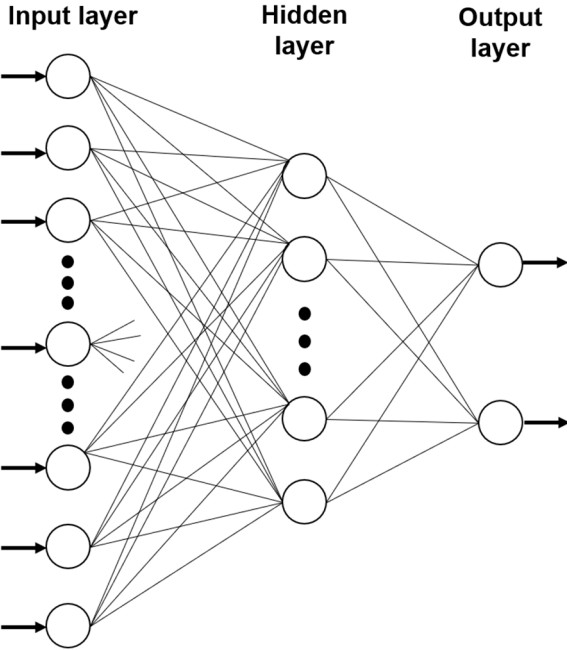

**Figure 4.** Feedforward neural network structure.

2.2.2. Recurrent Neural Networks

In recurrent neural networks (RNNs) the information goes back and forth as can be seen in Figure 5a, for this reason, they are also called feedback networks. In these networks, the connections between nodes form a directed cycle, where at least one path leads back to the initial neuron. In this type of network there are different types of structure [86]:

- Hopfield network: each neuron is completely symmetrically connected with all other neurons in the network. If the connections are trained using Hebbian learning, then the Hopfield network can function as a solid memory and resistant to the alteration of the connection. Hebbian learning involves synapses between neurons and their strengthening when neurons on both sides of the synapse (input and output) have highly correlated outputs [87] as shown in Figure 5b. There is a guarantee in terms of convergence for this network [88].
- Elman network: this is a horizontal network where a set of "context" neurons is added. In Figure 5c the context units are connected to the hidden network layer fixed with a weight. The subsequent fixed connections result in the context units always keeping a copy of the previous values of the hidden units, maintaining a state, which allows sequence prediction tasks [89].
- Jordan network: these are very similar to Elman's networks. However, context units feed on the output layer instead of the hidden layer.

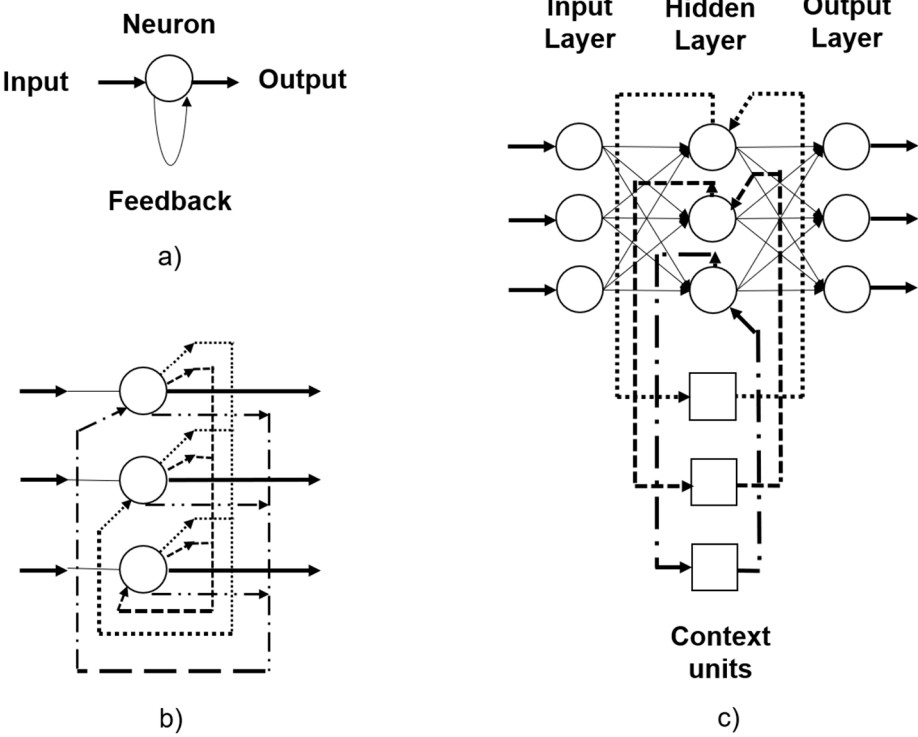

**Figure 5.** Recurrent neural networks structure: (**a**) Simple structure of a recurrent network, (**b**) Hopfield network structure, (**c**) Elman network structure.

RNN is distinguished from a FFNN by the presence of at least one feedback connection. FFNNs do not have the intrinsic ability to process temporary information. There are two important considerations about why recurrent networks are viable tools for modeling: inference and prediction in noisy environments. In a typical recurrent network architecture, the activation functions of the hidden unit are fed back each time step to provide additional input. That is, the recurrent networks are built in such a way that the outputs of some neurons feed back to the same neurons or to the neurons in the previous layers [86]. Feedback from hidden units allows filtered data from the previous period to be used as additional input in the current period. This causes the network to work not only with the new data, but also with the past history of all entries, as well as their leaked equivalents. This additional filtered input history information acts as an additional guide to assess the current noisy input and its signal component. By contrast, filtered history never enters a FFNN. This is where recurring networks differ from a FFNN. Second, since recurrent networks have the ability to maintain the past history of filtered entries as additional information in memory, a recurrent network has the ability to filter noise even when the noise distribution can vary over time. In a FFNN a completely new training must be carried out with a new data set containing the new type of noise structure [78].

### 2.3. Learning of Artificial Neural Networks

Learning is an essential part of NNs; this process defines the input-output relationship by looking for the most accurate prediction calculation. The learning process can be classified into two categories: supervised and unsupervised. Supervised learning knows the expected results and uses known or labeled data, while in unsupervised learning it is not necessary to have known data, and the learning is done through the discovery of internal structures and data representation [88,90].

Supervised learning consists of minimizing a cost function that accumulates the errors between the actual outputs of the system and the desired outputs, for the given inputs. To minimize this cost function, several methods are used, and the gradient descent as the error BP algorithm is the most used for its acceptable results in one layer and multilayer networks. [91].

In unsupervised learning, it is based only on input data and the update of the weights is carried out internally in the network, the algorithms are designed for the self-organization of the ANNs and can be derived by Hebbian law, or the use of algorithms such as algebraic reconstruction technique [68].

The exposed be learning techniques have allowed the development of advanced algorithms such as SOM (self-organizing maps) and SOTA (self-organizing tree algorithm), which are times series clustering algorithms based on unsupervised NNs [92]. SOM is a known data analysis tool for tasks like data visualization and clustering. One disadvantages of this tool is that the user must select the map size. This may lead to many experiments with different sized maps, trying to obtain the optimal result. Training and using these large maps may be quite slow [93]. While the SOTA permits classification in the initial levels of groups of patterns that are more separated from other and to classify patterns in final layers in a more accurate way [94]. These techniques open up the possibility of not only learning connection weights from examples, but also learning a neural network structure from examples. This is thanks to the fact that a neural network can be built automatically from the training data by SOTA methods [95].

## 3. Application of Artificial Neural Networks for the Prediction of the Greenhouse Microclimate

The application of methods and tools that simplify the treatment of the variables related to the climate of the greenhouses is a very important subject since the calculation speed, the precision in the prediction of the behavior and control of the variables of the different elements remain a significant challenge. ANNs are used to attend to these tasks largely by non-linear systems models [74]. Among the main studies that evaluated the viability of NNs in modeling the state of the greenhouse climate is [96], which focused specifically on the input-output relationships and the most efficient election process of inputs, although the training of the network was not an important part of their studies, proved that the ANNs obtain better results than the physical models of mass and energy transfer, and also emphasized their potential application for the environmental control of greenhouses.

### 3.1. Greenhouse Microclimate

Greenhouses are complex and non-linear systems, and a means to achieve a controlled agricultural production [7]. Greenhouse production systems have a complex dynamic impulse by external factors (meteorological), control mechanisms (ventilation openings, exhaust fans, heaters, evaporative cooling systems, etc.) and internal factors (crops and internal components) [6].

Concerning the greenhouse microclimate and its control, the crop represents the central element, but also the most complex part of the system. Due to this complexity and the great diversity of crops in greenhouses, it is common to consider only certain general issues that are more relevant to the response of crops in relation to greenhouse microclimate [97].

Greenhouse climates refer to the set of environmental variables in this system that affect the growth of crops and their development [98]. Greenhouse microclimate control has received considerable attention in recent years due to its great contribution to the improvement of crop yield [13,29,99]. The different factors such as temperature [100,101], relative humidity [102], amount of $CO_2$ [103,104] are analyzed to predict different events implementing artificial intelligence, statistics and engineering [49,105–108].

Numerous greenhouses use a conventional control, but this control strategy may not be suitable to guarantee the desired performance [15,26]. In this scenario, various strategies and control techniques have been proposed like generalized predictive control [109], optimal control [110], model predictive control [16,111], NNs control [112], fuzzy control [113–115], robust control [15,116] and linear-quadratic adaptive control [117]. The vast majority of these proposals are simulations of the behavior of the variables and possible control against these changes and are also focused on a specific crop [118].

The application of NNs in the control of microclimates is a topic that has currently gained interest. NNs provide reliable models that can reflect the non-linear characteristics of the greenhouse

that are difficult to solve using traditional techniques, do not require any prior knowledge of the system. and are very suitable for modeling dynamic systems in real time [119,120].

Temperature and humidity are of the most relevant parameters in the greenhouse microclimate, since they have complex exchanges and interactions of heat and mass between the inner air, other elements of the greenhouse and the outside. Building a model is a difficult assignment with simple mathematical formulas or transformation functions. However, the method of building models with ANNs has a great capacity for mapping non-linear functions, which is applied to many production process systems [121].

For the network design, air temperature and humidity of the greenhouse air are generally considered outputs, this due to the aforementioned factors. However, setting the inputs is more complicated and required an understanding of the system. It is not convenient to consider a large number of inputs, as this could cause uncontrolled extrapolations instead of increasing the estimation power. Three elements can be considered in order to consider an input variable: (1) correlation of selected input with other inputs, (2) physical dependence nature of the output and the input (3) input variable range. The third point, solar radiation and greenhouse temperature, can be considered an example [96].

### 3.2. Feedforward Neural Networks Models for Prediction of Microclimate in Greenhouse

Ferreira et al. [112] modeled the indoor temperature of a hydroponic greenhouse based on indoor relative humidity, outdoor air temperature, and solar radiation. They discuss different training methods for a neural network of radial base function (RBF) that are structurally simpler than multilayer perceptron (MLP), which are a type of FFNNs. The objective of using a radial base function artificial neural network (RBFANN) is that the design and training process is a simpler task. The training methods compared are the off-line and on-line, mainly differentiated in that the use of the learning algorithm adjusts the free network parameters as the output or input data are determined, respectively. In the study, they concluded that for both off-line and on-line training, better results are obtained by applying the Levenberg–Marquardt (LM) method, which is the best online. Other works applied the RBFANNs, as is the case of Hu et al. [99] who presented an adaptive proportional and derivative control (PD) scheme based on the RBF neural network. The RBF network used it to adjust the parameters of the PD controller using the Jacobian information for the greenhouse climate control problem. The results showed that the proposed adaptive controller obtained a more satisfactory performance than a conventional PD scheme and was even considered for application in non-linear dynamic systems such as the climate system of a greenhouse. Furthermore, Zeng et al. [38] presented a control strategy that combines RBF with PID, for greenhouse climate control. They compared the proposed adaptive online adjustment method with the offline adjustment scheme that uses GA to find the optimal gain parameters based on the error criteria. Offline learning consists on adjusting weight vectors and network thresholds after the entire training set is presented (requires at least one training data stage), while in online learning network weights and threshold adjustments are made after each training sample is submitted (after executing the adjustment step, the sample can be discarded) [122]. Interesting results were obtained such as better set point monitoring performance, a smoother control process characterized by smaller oscillatory amplitudes that the control can be applied in real time online and that the control scheme adapts well to fluctuations in external climate.

Regarding the MLP networks, which are a type of FFNNs, Dariouchy et al. [123] used them with training based on a gradient BP algorithm to predict the internal temperature and internal humidity within a tomato greenhouse from external climatic data (external humidity, total radiation, wind direction, wind speed, and external temperature). When comparing the results obtained from the network with a multiple linear regression method (MLR), the prediction of the MLP network proved to be significantly better. Also, He et al. [60] proposed a BP network based on principal component analysis (PCA) to predict the indoor humidity in a greenhouse. The PCA values were taken as inputs from the back propagation neural network (BPNN), the objective of the PCA was to simplify the data samples and make the model have a faster learning speed. The predicted humidity

coincided well with the measurement, which showed that the model had high accuracy. Furthermore, they compared the PCA-based BPNN method with a stepwise regression method and observed that the PCA-based BPNN performed better. Likewise, Taki et al. [124] used four MLP architectures with learning algorithms based on gradient descent momentum (GDM) and LM to predict roof temperature, indoor air humidity, soil temperature and soil moisture of a greenhouse. The results obtained showed that the prediction error is low and when compared with predictions obtained through regression models, the error used to predict the four parameters were approximately two times greater than the MLP method.

The structure of the network depends on the type of task to be described, the complexity of the system and the learning process. Table 2 shows various works in which FFNNs have been used, details the input and output variables that were used, the architecture of the network, the activation functions that were used by the network or each layer of the network, and the algorithm used in the training process.

**Table 2.** Applications of feedforward neural network models for prediction of microclimate in greenhouses.

| Author | Inputs Variables | Outputs Variables | Artificial neural network (ANN) Architecture | Activation Functions | Training Method | Comments |
|--------|------------------|-------------------|---------------------------------------------|---------------------|-----------------|----------|
| Zeng et al. [38]; Hu et al. [99] | • Outside temperature<br>• Outside humidity<br>• Wind speed<br>• Solar radiation<br>• Carbon dioxide concentration<br>• Heating<br>• Ventilation<br>• Carbon dioxide injection | • Inside temperature<br>• Inside humidity | • Feedforward neural network (FFNN) specifically radial base function (RBF).<br><br>The model had three layers:<br>• Input layer<br>• Hidden layer<br>• Hidden layer<br>• Output layer | • Gaussian transfer function for the hidden layer | Gradient descent back-propagation (BP) | Results show that the model proposed has better adaptability, and more satisfactory real-time control performance compared with the offline tuning scheme using genetic algorithm (GA) optimization and proportional, and derivative control (PD) method. |
| He et al. [60] | • Outside air temperature<br>• Outside humidity<br>• Wind speed<br>• Solar radiation<br>• Inside air temperature<br>• Open angle of top vent and side | • Inside humidity | *FFNN.*<br>The model had three layers:<br>• Input layer<br>• Hidden layer<br>• Hidden layer<br>• Output layer | • The sigmoid transfer function for the hidden layer<br>• The logistic sigmoid transfer function for the output layer | BP | The principal component analysis (PCA) simplified the data samples and made the model had faster learning speed. |

| | | | | | | |
|---|---|---|---|---|---|---|
| | • vent<br>• Open ration of sunshade curtain | | | | | |
| Ferreira et al. [112] | • Outside air temperature<br>• Solar radiation Inside humidity | • Inside temperature | *FFNN specifically RBF.* | | Off-line methodology:<br>• In method 1 they used the linear least squares (LS)<br>• In method 2 they used the orthogonal least squares (OLS)<br>• In method 3 they used the Levenberg – Marquardt (LM)<br>On-line methodology:<br>• In method 1 they used the extended Kalman filter (EKF)<br>• In method 2 they based on the interpolation problem with generalized radial basis functions (GRBFs) with regularization<br>• In method 3 they used the LM | In this paper off-line training methods and on-line learning algorithms are analyzed. Whether off-line or on-line, the LM method achieves the best results. |
| Dariouchy et al. [123] | • External humidity<br>• Total radiation<br>• Wind Direction<br>• Wind Velocity<br>• External Temperature<br>• Internal temperature<br>• Internal humidity | • Internal temperature<br>• Internal humidity | *FFNN.* | Logistic sigmoid transfer function for all layers | BP | Different architectures were tested. Initially, networks with a single hidden layer were built by successively adding two additional neurons to it. Networks with two hidden layers were also tested, triangular structures were considered, for which the number of neurons in one layer is greater than the next. The optimal model was composed of a hidden layer with six neurons. |
| Taki et al. | They used four | They used four | *FFNN.* | • Sigmoid | Basic BP | Demonstrated that multilayer |

| [124] | ANNs models:<br>First model:<br><ul><li>Inside air temperature</li><li>Solar radiation on the roof</li><li>Wind speed</li><li>Outside air temperature</li></ul>Second model:<br><ul><li>Inside soil temperature</li><li>Inside air humidity</li><li>Solar radiation on the roof</li><li>Inside air temperature</li></ul>Third model:<br><ul><li>Inside air temperature</li><li>Solar radiation on the roof</li><li>Inside roof temperature</li><li>Inside air humidity</li></ul> | ANNs models:<br>First model:<br><ul><li>Roof temperature</li></ul>Second model:<br><ul><li>Soil humidity</li></ul>Third model:<br><ul><li>Soil temperature</li></ul>Fourth model:<br><ul><li>Inside air humidity</li></ul> | | transfer function for the hidden layer<br><ul><li>Linear transfer function for the output layer</li></ul> | perceptron (MLP) network with 4 inputs in first layer, 6 neurons in hidden layer and one output, and MLP network with 4 inputs in the first layer, 9 neurons in hidden layer and one output had the best performance to predict inside soil, inside air humidity, inside roof and soil temperature with a low error. |
|---|---|---|---|---|---|

| | Fourth model:<br>• Inside air temperature<br>• Inside roof temperature<br>• Outside air temperature<br>• Solar radiation on the roof | | | | | |
|---|---|---|---|---|---|---|
| Seginer et al. [125] | Weather variables:<br>• Outside temperature<br>• Outside humidity<br>• Outside solar radiation<br>• Wind speed<br>Control variables:<br>▪ Heater heat flux<br>▪ Vent opening angle<br>▪ Misting time fraction<br>State variables:<br>▪ Leaf area | ▪ Inside temperature<br>▪ Soil temperature<br>▪ Inside humidity<br>▪ Inside radiation | *FFNN.*<br>For the model of the neural network (NN) used a commercial program (NeuroShell™, Ward System Group, Inc)<br>The model had three layers:<br>▪ Input layer<br>▪ Hidden layer (The number of the nodes was determined by the program)<br>▪ Output layer | Sigmoid function (S-shape logistic function) for the three layers | BP | They found that leaf area index (LAI) did not have a significant influence on the internal conditions of the greenhouse. Also, they determined that the wind direction has minimal effects on the results. |

| | | | | | | |
|---|---|---|---|---|---|---|
| | index (LAI)<br>Time variables:<br>▪ Julian day<br>▪ Hour of day | | | | | |
| Laribi et al. [126] | ▪ Outside temperature<br>▪ Outside humidity<br>▪ Wind speed<br>▪ Solar radiation | ▪ Internal temperature<br>▪ Internal humidity | *FFNN.*<br>The networks had three layers:<br>▪ Input layer<br>▪ Hidden layer with 7 neurons<br>▪ Output layer with 2 neurons | ▪ The sigmoid transfer function for the hidden layer<br>▪ The linear transfer function for the output layer | BP | Two approaches were used to predict the climate of the greenhouse, multimode modeling and neural networks. They point out that the neural network model is easier to obtain and specify that it can be used to simulate different output variables at the same time. |
| Bussab et al. [127] | ▪ External temperature<br>▪ External global radiation<br>▪ External relative humidity<br>▪ Wind speed | ▪ Internal Temperature<br>▪ Internal Relative Humidity | *FFNN.*<br>A multilayer NN with two hidden layers:<br>▪ First hidden layer with 40 neurons<br>▪ Second hidden layer<br>▪ with 20 neurons | ▪ The hyperbolic tangent function for input layer and for the first hidden layer<br>▪ The linear function for second hidden layer | BP | The NN obtained better results in the prediction of the internal temperature than of the internal relative humidity |
| Salazar et al. [128] | ▪ Outside average temperature<br>▪ Outside relative humidity<br>▪ Wind velocity | Three different network architectures were tested, where the number of outputs was varied:<br>• 1st inside | *FFNN.*<br>The networks had three layers:<br>▪ Input layer<br>▪ Hidden layer<br>▪ Output layer | Hyperbolic tangent function for all layers | BP | They report that the third network obtained better results in the prediction of temperature and relative humidity, which explains the interactions between these two variables. Also, they emphasize the relevance of the input variables in the |

| | | | | | | |
|---|---|---|---|---|---|---|
| | ▪ Solar radiation | • temperature • 2nd inside relative humidity • 3rd inside temperature and relative humidity | | | | predicted variables, in this study the solar radiation was the most important. |
| Alipour et al. [129] | ▪ Wind speed and direction ▪ Relative humidity ▪ Infra-red light ▪ Visible light ▪ Air temperature ▪ Carbon dioxide concentration | ▪ Inside temperature ▪ Light ▪ Inside Relative humidity ▪ Carbon dioxide | *FFNN.* Three different configurations were tested: ▪ The feedforward neural network with several delays in input ▪ Two layers with one feedback from the hidden layer and delay in input ▪ Three layers neural network with two feedbacks from hidden layer and delay in input | Not specified | | The three-layer neural network with two hidden-layer feedbacks and delayed entry showed better relative humidity and light index results. The FFNN with multiple entries delays better predicted the temperature and infrared index. |
| Outanoute et al. [130] | Values and the previous value of: ▪ External temperature ▪ External relative humidity ▪ Command of heater | ▪ Internal Temperature ▪ Internal relative humidity | *FFNN.* The networks had three layers: ▪ Input layer ▪ Hidden layer (the number of nodes depending on the type of network training) ▪ Output layer | ▪ The logistic sigmoid transfer function for the hidden layer ▪ The linear transfer function for the output layer | ▪ Gradient descent with momentum and adaptive learning rate algorithm (GDX) for seven nodes on the hidden layer ▪ Broyden-Fletcher-Golfarb-Shanno (BFGS) quasi-newton BP for five nodes on the hidden layer ▪ Resilient Back-propagation algorithm (RPROP) for twelve nodes on the hidden layer | Three NNs were tested with different training algorithms. BFGS is better than the GDX and the RPROP. |

| | | | | | | |
|---|---|---|---|---|---|---|
| | | and ventilator<br>Previous values of:<br>▪ Internal temperature<br>▪ Internal relative humidity | | | | |
| Taki et al. [131] | ▪ Outside air temperature<br>▪ Wind speed<br>▪ Outside solar radiation | ▪ Inside air temperature<br>▪ Soil temperature<br>▪ Plant temperatures | *FFNN.*<br>▪ Feedforward networks, specifically MLP and RBF, were used in this investigation. Also, different algorithms for network training were applied and compared with each other and with the support vector machine (SVM) method | For MLP:<br>▪ No transfer function for the first layer was used<br>▪ Sigmoid functions for the hidden layers<br>▪ The linear transfer function for the output layer<br>For radial base function artificial neural networks (RBFANNs):<br>Used radial basis functions as activation functions | ▪ LM back- propagation<br>▪ Bayesian regularization<br>▪ Scaled conjugate gradient BP<br>▪ RPROPVariable learning rate BP<br>▪ Gradient descent with momentum BP<br>▪ Gradient descent with adaptive learning rate BP<br>▪ Gradient descent BP<br>▪ BFGS quasi-Newton back-propagation<br>▪ Powell–Beale conjugate gradient BP<br>▪ Fletcher–Powell conjugate gradient BP<br>▪ Polak–Ribiere conjugate gradient BP<br>▪ One step secant BP | Thirteen different training algorithms were used for ANNs models. Comparison of the models showed that RBFANNs has lowest error between the other models |

### 3.3. Recurrent Neural Networks Models for Prediction of Microclimate in Greenhouses

Compared to FFNN, RNN application is a less studied field as can be seen in table Table 3, however, the results obtained show a good performance because they have a faster calculation due to the lower number of units in the input layer and a recovery structure similar to the structure of FFNNs training [132]. Fourati et al. [133] used an Elman-type RNN to simulate the dynamics of a greenhouse. In addition, for the control of the greenhouse, they developed an FFNN with a reverse learning process. In the operation and control of the greenhouse, they connected both NNs in cascaded obtaining a lower criterion error (Ec = 344.12) in comparison of a neural network simple (Ec = 533.31). Later, Fourati et al. [134] reapplied the same structure of the Elman-type RNN. However, they focused on developing a neuronal control strategy based on online training, adjusting the parameters of the controller (connection weights) with a generalized and specialized learning. That is to say, after offline learning, they applied the neuronal controller trained to provide control actions to the greenhouse through online learning. As a result, they obtained that the neuronal control took into account the new situations in the greenhouse environment by adjusting the aforementioned neuronal weights through online learning. On the other hand, Hongkang et al. [135] used an RNN model as a deep learning (DL) algorithm for predicting the temperature and humidity of a greenhouse. For the learning process, they applied the Elman type, they also applied the dynamic BP algorithm to modify the connection weights to reduce the prediction error and improve the learning capacity. When comparing this method with a BP network and an untrained RNN, they concluded that the Elman-type network based on the BP dynamic algorithm can predict temperature and humidity accurately and in the short term in the next step based on the data of the interior environment due to recursive online training. They also mention that the use of an RNN for the prediction and control of climate in greenhouses is very effective since the online model can adapt to changes and guarantee its evolution.

**Table 3.** Applications of recurrent neural network models for prediction of microclimate in greenhouses.

| Author(s) | Inputs Variables | Outputs Variables | Artificial neural network (ANN) Architecture | Activation Functions | Training Method | Comments |
|---|---|---|---|---|---|---|
| Fourati et al. [133] | ▪ External temperature<br>▪ External hygrometry<br>▪ Global radiant<br>▪ Wind speed | ▪ Internal temperature<br>▪ Internal hygrometry | *Recurrent neural networks (RNN).*<br>▪ Elman neural network | Sigmoid function for the hidden layer | Back-propagation (BP) | Elman neural network was used to emulate the direct dynamics of the greenhouse. Based on this model, a multilayer feedforward neural network (FFNN) was trained to learn the inverse dynamics of the process to be controlled. |
| Fourati et al. [134] | ▪ External temperature<br>▪ External hygrometry<br>▪ Global radiant<br>Wind speed | ▪ Internal temperature<br>▪ Internal humidity | *RNN.*<br>▪ Elman neural network | Sigmoid function for the hidden layer | Neural control using with Online training:<br>▪ Generalized learning<br>▪ Specialized learning | In order to evaluate the different control strategies (offline and online training), they defined an error criterion. When they compared the error between training methods, obtained that online methods are better than offline method (FFNN based on Elman neural network). |
| Hongkang et al. [135] | | ▪ Internal temperature<br>▪ Internal humidity | *RNN.*<br>▪ Elman neural network | Sigmoid function for the hidden layer | Dynamic BP | Different from the traditional batch trained neural network, the dynamic BP method in the training process uses the output of the previous step together with the next input to the network, and the |

| | | | | | | |
|---|---|---|---|---|---|---|
| | | | | | | calculator outputs the weights. They compared a dynamic BP RNN whit untrained RNN, the Elman network based on dynamic BP algorithm can accurately predict the temperature and humidity in the greenhouse better than the untrained RNN |
| Dahmani et al. [136] | ▪ External temperature<br>▪ External Humidity<br>▪ Global radiation<br>▪ Wind speed<br>▪ Heating input<br>▪ Opening of the shutter<br>▪ Misting input<br>▪ Curtain entrance | ▪ Internal temperature<br>▪ Internal humidity | *RNN.*<br>▪ Elman neural network | Sigmoid function for the hidden layer | BP | The control law is based on a multilayer perceptron (MLP) network type trained to imitate the inverse dynamics of a greenhouse. The direct dynamics of the greenhouse were described by a RNN of the Elman type |
| Salah et al. [137] | ▪ External temperature<br>▪ External hygrometry<br>▪ Heating<br>▪ Sliding shutter in degrees<br>▪ Sprayer<br>▪ Curtain | ▪ Internal temperature<br>▪ Internal hygrometry | *RNN.*<br>Three Elman neural network are considered:<br>▪ One hidden and context layers<br>▪ Two hidden and context layers<br>▪ Three hidden and context layers | Sigmoid function for the hidden and output layers | Deep learning (DL) where BP algorithm was used | Concluded that the network with two hidden layers and two context layers were the most efficient to describe the system |

*3.4. Other Artificial Neural Networks Models for Prediction of Microclimate in Greenhouses*

Other models using ANNs have also been developed as shown in Table 4. Rodríguez et al. [138] built a neural network model in a non-linear autoregressive configuration (NNARX). The model was carried out through NNs s using the lagged values of the measurable signals as input vector. The structure of the network consisted of an MLP, concluding that the network for long-range prediction purposes does not guarantee optimal results, with one step or two step prediction being more convenient, but to make a long-range prediction they recommend decreasing the number of autoregressive entries and increasing the number of non-regressive entries. Similarly, Manonmani et al. [139] developed intelligent control schemes based on a NNARX to control the internal temperature and humidity of a greenhouse. In their work, they propose two intelligent control schemes, a neural predictive controller (NPC) and a non-linear autoregressive mobile average controller (NARMA-L2).

The results of the temperature and humidity simulation indicated that the NARMA-L2 controller provides good monitoring of the setpoint and disturbance rejection capabilities. The performance indices showed that the setup time is shorter for the NARMA-L2 controller than for the NPC. They also found that unlike other ANN control schemes, the NARMA-L2 controller uses only a neural network for modeling and control. Models have been made to predict the internal temperature of the greenhouse using an auto-regressive model with external input and neural networks (NNARX) as proposed by Frausto et al. [5], the network structure consisted of an MLP where the entrance to this structure was using a vector containing the regressors of an auto regressive with exogenous input (ARX) model and the training was using the BP algorithm. The models showed good performance (daily average absolute simulation error smaller than 1 °C) for long periods without the need to readjust the parameters frequently, they also indicate that the number of neurons in the hidden layer of the NNARX system plays an important role in obtaining a good performance.

**Table 4.** Other types of neural network models for prediction of microclimate in greenhouses.

| Author(s) | Inputs variables | Outputs variables | Artificial neural network (ANN) Architecture | Activation functions | Training method | Comments |
|---|---|---|---|---|---|---|
| Lu et al. [140] | ▪ External temperature<br>▪ External humidity<br>▪ Internal temperature<br>▪ Internal humidity | ▪ Internal temperature<br>▪ Internal humidity | *Nonlinear autoregressive with external input neural network (NNARX)*<br><br>The fundamental structure was three-layer feedforward neural network (FFNN):<br>▪ Input layer with 2 nodes<br>▪ Hidden layer with 2 neurons<br>▪ Output layer with 1 neuron | ▪ Hyperbolic tangent function for hidden layer<br>▪ Linear transfer function for the output layer | Levenberg–Marquardt (LM) | Compared the NNARX with the genetic algorithm (GA) model, the prediction obtained by the neural network (NN) method was better |
| Zhang et al. [141] | ▪ Temperature<br>▪ Humidity | ▪ Skylight<br>▪ Sun-shade net Circulation fan<br>▪ Side windows<br>▪ Fuel heater<br>▪ Micro-mist humidifier | *Fuzzy Neural Network*<br><br>The structure was four-layers:<br>▪ Input layer<br>▪ Second layer were represented a linguistic variable<br>▪ Third layer where the | The inputs and outputs are fuzzified | Gaussian function as the membership function for the layers | Compared the fuzzy neural network controller with the conventional proportional, integral and derivative controller (PID) to verify the performance. The fuzzy neural network had small overshoot, fast |

| | | | | | | |
|---|---|---|---|---|---|---|
| | | | function was to complete the fuzzy logic inference, and calculate the fitness of each rule<br>▪ Output layer | | | response, good stability, and small steady-state error |
| Patil et al. [142] | ▪ Outside air temperature<br>▪ Outside air relative humidity<br>▪ Global solar radiation flux density<br>▪ Cloud cover | ▪ Inside air temperature | *NNARX.*<br><br>The fundamental structure was three-layer feedforward neural network:<br>▪ Input layer with 4 inputs<br>▪ Hidden layer with 24 neurons<br>▪ Output layer with one output | ▪ Hyperbolic tangent function for hidden layer<br>▪ Linear transfer function for the output layer | LM | Eighteen different models were tested. auto regressive with exogenous input (ARX), autoregressive moving average with exogenous input variables (ARMAX) and NNARX models were compared to each other and concluded that NNARX performed better. |

## 4. Artificial Neural Networks in Energy Optimization of Greenhouses

Another problem of interest in greenhouses is the optimization of energy consumption derived mainly from heating and, ventilation systems, among other control elements [143]. Optimal control strategies, for the most part, are based on mathematical models for calculating greenhouse energy consumption and mathematical methods to minimize total energy consumption. An example is the state energy balance model's use. The use of this model is not new [144], nor is its use for real-time energy optimization [9]. However, the implementation of these techniques with sustainable technologies such as photovoltaic (PV) collectors have allowed predicting performance and establishing better systems for energy consumption [145]. Furthermore, greenhouses with systems that optimize energy consumption must assess heating needs before being implemented, and one of the ways to do this is through mass flow and energy transfer models [146]. Currently these models are still being developed to predict heating requirements. They make it possible to resolve different issues related to this topic, such as the forecast of the hourly energy requirements based on the entry of the parameters of environmental control inside the greenhouse, the physical and thermal properties of the crops and the construction materials [147]. In addition, CFD-based energy saving and system performance models have been proposed [148].

The use of other types of models such as based optimization techniques such as particle swarm optimization (PSO)and GA have also shown good results [149], as well as with NNs [10]. Energy consumption is largely derived from two factors that influence the aforementioned control elements, temperature, and humidity. Trejo-Perea et al. [150] developed a predictor of energy consumption for greenhouses from an MLP, also compared the ANN model with a non-linear regression model. The results obtained show that the prediction power of the network is superior to the regression model with a significant accuracy level (95%). Regarding the structure of the network, a cascade architecture was carried out where the input variables were temperature, relative humidity, time and electrical consumption, on the other hand, the output variable considered was the electrical consumption. Several MLP models were tested, where the hidden layer was the only variant with five, four, three and two neurons. While the Levenberg–Marquardt reverse propagation algorithm was used for the learning procedure. The MLP model with the best results was the model with three nodes in the hidden layer, also compared to the regression model.

The use of elements that help the energetic production in greenhouses is also a topic of interest, in the same way, its energy management and optimization. Photovoltaic modules are a viable option for this task, Pérez-Alonso et al. [151] developed a photovoltaic greenhouse, where the use of ANNs focused on the prediction of instantaneous production of the system. The network used was feedforward trained using an LM algorithm. The input variables considered were ambient temperature, relative humidity, wind speed, wind direction, and radiation. As output variables, only photovoltaic energy production was considered. The hidden layer of the network consisted of 140 neurons, the tests were obtained in 1 second and prediction errors for the instantaneous production of electricity below 20 Watts.

Other studies have used ANNs to predict greenhouse production using the amount of energy use as a basis. Such is the case of Taki et al. [152] who, through an MLP network, predicted greenhouse tomato production. They used as inputs the energy equivalences of chemical products, human energy, machinery, chemical fertilizers, diesel fuel, electricity, and irrigation water. The architecture used consisted of 7 inputs, 10 neurons for each of the two hidden layers and one output (tomato production). No transfer function was used for the input layer, for the hidden layers a hyperbolic tangent transfer function was used, and for the output layer a linear transfer function was chosen. The results revealed that diesel fuel (40%), chemical fertilizers (30%), electricity (12%) and human energy (10%) consumed most of the energy. The comparison between the ANN model and the multiple linear regression model (MLR) showed that the ANN model predicts the output performance significantly better than the multiple MLR model.

Development of new control strategies influence energy costs by reducing the energy consumption of greenhouses. However, the potential for energy saving control seems to be

over-estimated. Climate control strategies for energy saving have been developed [153], from the analysis of greenhouse roofing materials and how these affect energy consumption [154] to the use of thermal screens and how they can reduce consumption of energy at night [155]. Likewise, the response of the crop has been investigated when applying techniques for energy saving [156], however, it is necessary to explore more methods beyond those exposed and the NNs application as a viable tool.

## 5. Other Applications of Artificial Neural Networks in Greenhouses

The ability of ANNs to model complex and non-linear systems allows their application in different tasks in greenhouses, not only in predicting the microclimate where the great majority of studies focus. As indicated, the internal temperature and humidity are among the variables that generate the most interest to predict their behavior. However, other elements such as $CO_2$ enrichment in hot climates exert considerable weight for the proper functioning of the greenhouses, since a balance is required between the need to ventilate and enrich as explained by Linker et al. [157]. They developed NNs for the prediction of temperature and $CO_2$ concentration separately, the training algorithm used was the BP. The activation function chosen for the hidden layer was sigmoidal, while the linear activation function was used for the output layer. In this case, it was decided to reduce the size of the NN instead of a more complex NN with multiple inputs and multiple outputs (MIMO).

The models fit the data well, and also generated reliable optimization results. In addition, they demonstrated the effect of evaporation cooling by extending the duration of $CO_2$ enrichment. Another aspect that is related to the concentration of $CO_2$ is photosynthetic efficiency and crop growth, and Moon et al. [158] performed an ANN to predict the concentration of $CO_2$ in greenhouses considered environmental factors. The network consisted of a feedforward, with an architecture of an input layer (10 neurons), two hidden layers (the number of neurons of 32, 64, 128, 256, 512, 1024, and 2048 were being changed with the aim of finding the optimal ANN, both layers had the same number of neurons) and one output layer (one neuron). The variables considered as inputs were internal temperature, internal relative humidity, internal atmospheric pressure, photosynthetic photon flow density (PPFD), external temperature, external relative humidity, external atmospheric pressure, wind speed, and wind direction while the $CO_2$ concentration was the output variable. The transfer function that was used throughout the layers was the rectified linear unit (ReLU) and the training algorithm was the AdamOptimizer. The results obtained show that the prediction of $CO_2$ concentration is possible through ANNs with a coefficient of determination of 0.97. However, the estimates made in the study were limited to data obtained from each greenhouse and the authors indicate that it is necessary that ANNs should be trained with data from several measurement sites to generalize all possible situations.

The networks potential for the growth improvement of greenhouse crops by means the forecast and description of the microclimate has been exposed, these studies are based on the fact of having the ideal conditions of the plant controlling through predictions one, two or several environmental factors. However, other research has evaluated the close relationship between crop yield, growth and water use in response to changes in the greenhouse climate. An ANN was developed by [159] to predict the yield, growth and amount of water used in tomato crops under the greenhouse. The input variables were radiated, relative humidity, growth, $CO_2$ concentration, and temperature. The network was a feedforward, they used software (Predict®, v3.21) for the construction of the network which was responsible for defining the structure and the training process. The yield, growth, and use of water responded similarly to the climatic variables. Radiation and temperature remain the most influential variables, however, the $CO_2$ concentration has a significant weight in the positive change of the output variables. On the other hand, Juan et al. [160] modeled the tomato growth process. The factors considered influential and input elements were solar radiation, temperature, humidity, and $CO_2$. A modified Elman network was used to model the dynamics of the system. They made arbitrary connections from the hidden layer to the context layer, they also used the hyperbolic tangent function as an activation function in the hidden layer, while in the other layers they used

linear activation functions. A fuzzy GA, was used for the learning process, which deals with a modification to the traditional method of GA through a crossover with fuzzy logic. The simulation results showed that the modified Elman network and the fuzzy genetic algorithm are better for the description of the system compared to an Elman network trained using a BP algorithm.

The transpiration of plants in greenhouses is an element that represents a challenge in matters of modeling since the elements that intervene with this phenomenon remain a challenge for their mathematical representation [161]. The application of ANNs for modeling the transpiration of greenhouse crops is a way of presenting reliable results, as presented by [162]. The exposed model consisted of a modified BP network, since the randomness of the conventional BP algorithm in the weights and the threshold in each training represented a disadvantage for the prediction of transpiration. The modified algorithm was a genetic algorithm that, through an optimization adjustment function selected the best weights and thresholds, used a network called genetic algorithms-back propagation neural network (GA-BPNN). Also, using a NNARX model, the error accumulated by the long training time was only recorded.

Wireless sensor networks (WSN) are a new form of distributed computing and are encompassing a wide variety of applications that can be implemented with them [163]. In greenhouses it is primarily concerned with collecting environmental information and sending it to the grouping nodes via wireless data link. WSN is a type of self-organizing wireless network that takes data as its core [164]. The role of this technology and ANNs is that they are a good combination for controlling greenhouses. WSN can be used to monitor $CO_2$ concentration [165]. Zhang et al. [166] carried out a greenhouse control system using a WSN to collect data on temperature, humidity and $CO_2$ concentration. They related the internal environmental factors and the actuators of the system for the implementation of a fuzzy rule and combined with a neural network. The fuzzy neural network consisted of three inputs and six outputs to improve control precision. Moreover, Ting et al. [167] measured and collected real-time data on air temperature, humidity, $CO_2$ concentration, soil temperature, soil moisture, and light intensity using WSN. The measurement of these parameters was to predict the photosynthetic rate of plants and in turn to quantitatively regulate $CO_2$. The prediction model was established based on a BP neural network. The environmental parameters were used as input neurons after being processed by PCA, and the photosynthetic rate was taken as the output neuron.

There are many important areas where WSN can improve. One of the aspects to consider is to give the sensor networks the ability to reprogram themselves wirelessly, allowing users not to physically interact with the sensor nodes. This wireless reprogramming can be based on the concept of NNs as proposed by Cañete et al. [163], and thus be able to implement it in greenhouses.

## 6. Perspectives: Greenhouse Artificial Neural Networks Application

### 6.1. Agriculture 4.0 and the ANNs

Farmers today need to adapt to new technologies and apply them in agriculture. Agriculture has gone through different stages, starting with agriculture 1.0 characterized by the use of animal force; then came agriculture 2.0 that used combustion engine machinery; moving to agriculture 3.0 where guide systems (such as geographic positioning (GPS) systems) and precision agriculture (PA) would be used; and finally agriculture 4.0, which is based on the principle that activities are connected to the cloud [168].

Agriculture 4.0 is the integration of technologies (IoT, PA, artificial intelligence (AI), cloud computing (CC), among others) through the cloud to automate cyber-physical tasks and systems, allowing the planning and control of production [169]. This new era arose when telematics and data management were combined with the concept of PA and largely driven by the use of the IoT [170]. PA is the management of spatial and temporal variation in fields with respect to soil, atmosphere, and plants using information and communication technologies. Its concept was born from the need for the development of site-specific techniques. In other words, it applies treatments to areas within

a field that requires different management than the field average, allowing fine-tuning of crop management systems [171].

IoT in the agricultural context refers to the use of sensors and other devices to convert every element and action involved in agriculture into data. IoT technologies are one of the reasons why agriculture 4.0 can generate such a valuable amount of information [172].

Agriculture based on agricultural data is known by different names apart from agriculture 4.0: Digital agriculture or SA. However, SA emerges as a main concept of agriculture 4.0. The SA addresses important agricultural objectives such as saving water, conserving the soil, limiting carbon emissions and increasing productivity by doing more without stopping [173].

### 6.1.1. Precision Agriculture and Internet of Things

The PA integrates the new technologies derived from the information age with the agricultural industry. It consists of a crop management system that tries to optimize the type and quantity of inputs with the real needs of crops for small areas within an agricultural field. PA uses crop inputs more effectively, including fertilizers, pesticides, tillage, and irrigation water [174].

As a management tool, PA consists of five elements: geographic positioning (GPS), information gathering, decision support, variable-rate treatment, and performance mapping. Yield mapping allows the farmer to monitor the actual result of the different inputs, being a tool for collecting information on previous years. For this reason, large data set (big data) are required to interpret specific variables. In this area, new technologies are still under development [175]. Mapping many different factors of soil, crops, and the environment produces large amounts of data. Farmer data overload must be overcome by integrating expert systems and decision support systems [176], which in turn must be based on models such as those that have been exposed throughout this paper.

PA has been applied and developed in greenhouses [177–179], as well as the use of NNs as a support tool [180]. Being the real-time monitoring systems for the management of the greenhouse to control environmental parameters, this is the area in which it is necessary to go deeper [181]. Likewise, SA broadens the concept of PA, since the tasks for decision management are reinforced by knowledge of the situation. This in turn causes real-time assistance resources to be required to perform agile actions such as the IoT [182].

IoT is the interaction between a variety of physical things or objects that use specific addressing schemes to connect to the internet, and this type of technology allows the inherent reduction of environmental impact by real-time reaction to alert events such as detections of weeds, pests or diseases, climate or soil monitoring warnings, which allow a reduction and the adequate use of inputs such as agrochemicals or water [183,184].

One of the advantages of IoT is its ability to control other devices remotely transversely based on the existing system, which makes a good interrelation between the physical world and different computer-based frameworks and creates possibilities for greater financial effectiveness advantages and precision. In the near future, IoT will be trusted with numerous administrative functions [185]. Currently, IoT has been implemented in crop care. Kitpo et al. [186] applied IoT to determine the date of tomato harvest, for this they carried out a monitoring of the 6 different stages of tomato cultivation, using as parameter the visible wavelength as a characteristic in the classification of support vector machines (SVM). Climatic data such as temperature, humidity, illuminance, among others, were recorded daily during tomato cultivation, these data and the data obtained from the SVM classification were used for the training of a NN, the results applied to the elaboration of an automated system by using IoT to support greenhouse growers in the future.

Tervonen [187] studied the effectiveness of IoT in quality control during vegetable storage. During the storage of potatoes, it determined that for the proper control of temperature and other parameters, multiple measurement points are required in different locations to guarantee the desired behaviors for the entire volume. Wang et al. [188] verified that the data loss rate between the data acquisition unit and the gateway was 1.52%, and the data loss rate was 0.4% between the gateway and the server, making the IoT system feasible for monitoring greenhouses. IoT has emerged as an alternative for optimizing the agricultural sector since it allows farmers to monitor

their agricultural fields in real time and receive recommendations to produce good quality crops while maximizing their overall profits on the products sold [189]. Linked to the IoT, there is the CC. CC is a model that allows convenient access to the network request to share configurable calculation resource groups [190], it is a model to allow ubiquitous, convenient network access and on demand to a shared pool of computing resources that can be quickly provisioned and released with minimal effort from management or service provider interaction [191].

### 6.1.2. Smart Agriculture

SA and PA are booming, but they could take advantage of technology in the agricultural world. SA is an agricultural management concept that uses modern technology to increase the quantity and quality of products, access to GPS, soil scanning, data management, and IoT technologies. In the case of smart greenhouses, evaluation of production, energy loss and increased labor costs is essential as a result of manual intervention against environmental impact. In addition, to control the climate, monitoring must be intelligent so that there is no need for manual intervention. The parameters necessary for efficient product production are determined by various sensors and the data are transferred to a cloud-based environment for evaluation [192].

One of the main disadvantages of the current agricultural greenhouses is the efficient and intelligent information management. That is, what is needed for the efficient implementation of technologies such as IoT is the design and implementation of the general system as shown in Figure 6. The design of these systems where internet or local area network (LAN) technology is used will allow sensors, controllers and computers to be combined to connect people and "things", thus obtaining data, and remote control and intelligent network management [193].

In traditional agriculture, pesticides, fertilization and irrigation depend on the experiencing of farmers; however, it does not guarantee the accuracy of parameters such as temperature, humidity, lighting and other indicators that are difficult to determine and adjust only by experience. In a smart greenhouse, by having a large number of sensors, the collected data can be communicated via the Internet and, therefore, to an operator. The operator might also have an Internet interface to control fertilization, irrigation, heating, lighting and other parameters [194].

The amount of information generated can be used to develop more robust models that predict the behavior of greenhouse parameters, and thus speak of an adaptable AI with more complex learning capacity. The challenge is that these models coexist with technologies such as CC and IoT. The ANNs, specifically deep NNs, which are powerful tools for prediction and optimization are an option for various applications in agriculture [137] and greenhouse agriculture. CNNs, RNNs and long short-term memory neural network can consider various types of information and handle a large amount of data, the potential of having information stored in the cloud and access to it from anywhere would allow training and use of networks to be more efficient, since any unit with computing capacity could use this information simultaneously and not only depend on each neural network, NNs could operate in parallel.

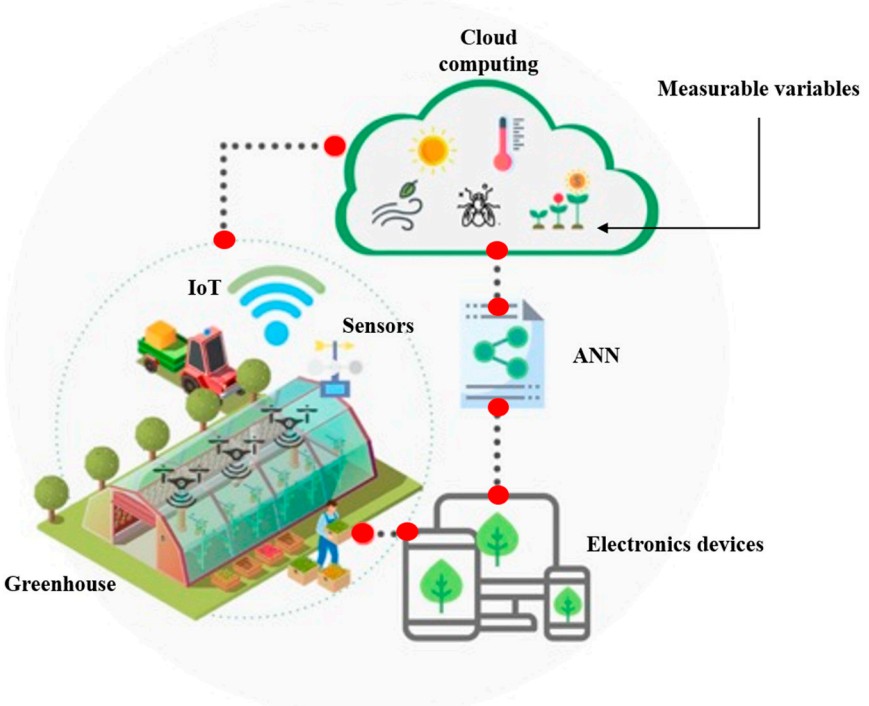

**Figure 6.** Agriculture 4.0 applied in a greenhouse.

*6.2. Artificial Neural Networks and Greenhouses*

The main topic involved in the development of the research is the efficient production of crops with the help of greenhouses to meet the growing needs and demands. That is, greenhouse cultivation can be an option to overcome such problems, problems that go hand in hand with economic development, ecology, and climatic conditions. Traditional greenhouses have changed in such a way that they can now be equipped with temperature, light, carbon dioxide, and relative humidity control systems. The optimization and adaptability to the changes that the system undergoes are vital to achieve an improved plant growth. These changes are in parameters such as temperature, water vapor, air pressure, air velocity, radiation rate, etc. In addition to this, greenhouses require a continuous supply of energy from renewable or non-renewable sources to maintain the internal microclimate with the aforementioned parameters [195].

The use of mathematical models that allow the prediction of changes and adaptability of the greenhouse has been thoroughly studied. The complexity of these models is given by the complexity of the greenhouse itself, the choice of which type of model (physical or those that analyze the inputs and outputs of the system) is the most convenient depending largely on time, resources, type of crop and type of greenhouse you have to implement it. The use of ANNs as an option to satisfy these demands has been developed for approximately 40 years. After analyzing 35 works, 74% focus on the description and prediction of the microclimate, 9% on energy optimization and 17% on other applications of greenhouse networks. The most used type of NN is feedforward with 46% of the investigations, while RNNs represent 20% and other types of NNs 32%. Figure 7 presents the topic in which NNs are used, as well as the most used architectures and the predominant training method.

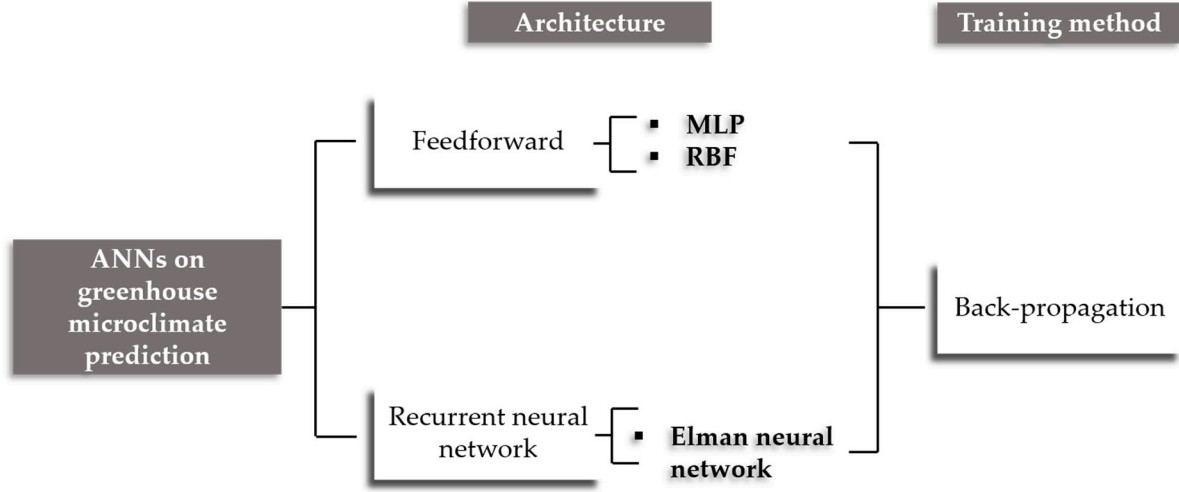

**Figure 7.** Artificial neural networks on greenhouse microclimate prediction. ANNs: Artificial neural networks; MLP: Multilayer perceptron; RBF: Radial basis function

Although the vast majority of the works present favorable results in the use of these types of models, there are several issues that need to be pointed out. The results are simulations performed for the validation generating models [38,60,99,135], where the implementation of the model in conjunction with the greenhouse control systems and the yield obtained compared to conventional systems are not presented. The construction of the networks in greenhouses has been carried out with cultivation [123,128,159,162] or without cultivation [129–131,134], however, the results obtained are not tested in the opposite situation is obtained with a greenhouse without cultivation is not tested with cultivation and vice versa. In addition, the models obtained can hardly be used in another type of greenhouse, so the generalization of the models should be a more relevant issue [125].

The importance of addressing the points described above is being able to apply ANNs daily in greenhouse production and integrate them to emerging technologies and make the change to agriculture 4.0 and SA as well. Ensuring that connectivity and data transmission are more efficient and economical. Work with the automation of knowledge work through models such as ANNs to manage assets and optimize the performance of the greenhouse production process by having improved sensors and remote monitoring; to implement CC where the integration of the measurement systems of greenhouses is done through the Internet; and, in addition, to improve artificial intelligence to automate precise tasks in this type of system [196]. Similarly, it can be used with the IoT for the design of new methods to solve problems in market demand, precision in operation and supervision [197]. In addition, the use of learning algorithms and activation functions to open the landscape in ML and provide powerful analytical tools will help establish more efficient control and automation systems in greenhouses. Of the papers presented, only [137] presented the option of deep learning, while Wang et al. [162] they explored the feasibility of using GA as an optimization resource to work together with NNs (the development of the GA-BPNN model). DL and hybrid NNs are rarely used in greenhouse agriculture; however, they are concepts that are currently being developed in different areas and can be the way for the development of networks that match the new technology and challenges faces agriculture in greenhouses.

### 6.3. Classic Models versus ANNs

Greenhouse climate models can be classified into two categories [125]: models for the design of new greenhouses and models for climate control of existing structures. The latter are also known as classic models, and they are based on steady state energy balances. The number of parameters in this type of model is small compared to MIMO (multiple input multiple output) black box models, this being one of its main advantages.

On the other hand, mechanistic models provide a clear physical explanation of the greenhouse environment such as static [198,199] and dynamic models [100]. Static models are based on the static energy balance of the greenhouse components and usually their heat storage capacities are not considered [200]. The relevance of physics-based models in greenhouses is that they take the physical parameters that describe the system, they can include the location of the greenhouse, local weather conditions, geometry, construction materials, hours of operation, systems of air conditioning and settings. That is, they allow its use for the design phase and help to evaluate the energy performance of the greenhouse [201,125]. However, the current state of climate control still leaves much room for improvement [202] and optimal control of greenhouse environments can be improved by combined models to allow selection of greenhouse designs and control algorithms to maximize the room for improvement benefit such as models based multi-objective optimization [203,204].

From the studies presented in Table 2, Bussab et al. [127] found that the efficiency of the FFNNs in models based on mass flow and energy equations is better when forecasting the relative humidity and internal temperature of a greenhouse. The ANN was more accurate in 81% of the cases than the classic method in forecasting the internal relative humidity and 62% more efficient in forecasting the internal temperature. These results comply with what was mentioned in Seginer et al. [125]. Among the reasons why ANNs are more efficient than classic models is that mentioned above, the ability to consider more parameters, that is, with a sufficient number of adjustable parameters, is capable of making accurate predictions, provided that it presents all the factors that have a significant influence on the outputs (in this case, the internal relative humidity and the internal temperature). In addition, in the case of ANNs, you can always choose to increase the number of neurons in the hidden layers to increase the predictive power of the network. However, there is a point at which the network will not show significant improvement no matter how many neurons it has in the hidden layer. For Bussab et al. [127], a configuration with two hidden layers was optimal, where the first hidden layer consisted of 40 neurons and the second of 20. Studies comparing the effectiveness of ANNs with classic models are few, however, Seginer et al. [125], Seginer [96] and Linker and Seginer [205] expose several factors that make networks have greater predictive power, the importance of reducing the number of inputs on a neural network and how it can help the network with classic models.

Supporting the ANNs with classic models for predicting the microclimate in greenhouses brings great benefits. Linker and Seginer [205] were among the first to develop such a model. The reason for using physical models in black box models, specifically in NNs is due to its main disadvantage. The poor extrapolation property, in other words, lacks prior knowledge, and the most evident in sigmoidal black box models. Linker and Seginer [205] demonstrated that hybrid models of this type produce efficient predictions, especially in the operational domain, decreasing their precision in the training domain. The proposal is to use the classical model to generate "synthetic" training data. In this way, prior knowledge of NNs can be included, solving the problem previously exposed. The configuration of the physical model can be done in two ways: serial and parallel. The one proposed in his work was in parallel, where synthetic data was generated for the training period for all the inputs during a two-year period. Experimental data from the training period was added to the synthetic database, and all synthetic points associated with the nodes for which at least one experimental data point was already available were removed. In this way, the synthetic data was only used in regions where no experimental data was available. The main problem with this technique is that the database can be very large and there is a risk that the experimental data may be lost among the synthetic data.

The use of physical models in conjunction with the FFANNs was also presented in other works, but with a totally different approach to that of Linker and Seginer. Hu et al. [99] proposed an RBF network to adjust the parameters of a conventional PD controller, they used the model based on energy equations and mass fluxes to address the humidity and temperature of the indoor air of a greenhouse. They compared this model against a model based on conventional RBF networks. When calculating the mean errors, they found that the proposed model had smaller values, also the control

signals were smoother. Likewise, in comparison with the previous models, the operation of the model was tested in conjunction with a control scheme, obtaining the decrease of the serious oscillation in the greenhouse actuators. Later, Zeng et al. [38] compared this same model with a method that uses GA. They found results that would be interesting to see in future studies with other models than GA. The main disadvantage of GA is that being an offline model it could not adjust to the external climate fluctuation of the greenhouse. Changes in solar radiation, external temperature, and external humidity caused a decrease in control performance, in addition, it is time consuming and depends on the GA optimization calculation time, and in practice its application in a real-time control system is not very convenient. By establishing the following advantages of a physical model-based RBF network over offline models such as GAs: they have better setpoint tracking performance, they have a smoother control process characterized by smaller oscillatory amplitudes, it can apply to real-time monitoring and, most importantly, it is well adapted to external weather fluctuations.

*6.4. The Input Variables in the ANNs and in the Prediction of Greenhouse Microclimate*

One of the important issues in a neural network is to reduce the number of inputs. He et al. [60] used PCA as the base in a BP neural network. In the proposed network, the input layer consisted of 4 inputs, which were the main components and the output layer was 1, the internal humidity of the greenhouse. To determine the behavior of the PCA-based BP network, it was compared to a conventional BPNN network, where the training times of these methods were 42 and 130, respectively. The PCA allows network training to be faster. However, although the PCA-based network achieved 85% accuracy it is still less than the conventional BPNN network, and this is because there is a loss of original data information. With these results they also compared the PCA-based BPNN network with a stepwise regression model, when calculating the mean squared error (MSE), the PCA-based BP network performed better with a value of 1.6745 while with the stepwise regression method they obtained a value of 4.5437. The use of techniques such as the PCA to determine and simplify the entries of the ANNs is a very viable option, although this may affect its effectiveness, it is still better than other methods and is rarely used.

Another way to delimit their number is to apply the sensitivity analysis to the different input variables to determine which ones are more relevant to the variables whose behavior is to be determined, just as did Seginer et al. [125], who found that solar radiation and outdoor air temperature are the factors that have the greatest impact on the temperature and internal humidity of the greenhouse. Similarly, Salazar et al. [128], using a sensitivity analysis, determined that the most important variables for predicting temperature are outdoor temperature and solar radiation. Both Seginer et al. [125] and Salazar et al. [128] analyzed the effect of considering separate models for each of the outputs, that is, a model to predict humidity, one more to predict internal temperature and finally a model in which consider two outputs, the humidity and the internal temperature of the greenhouse. They concluded that the separate models present poorer predictions, this due to noise from unnecessary inputs. For their part, Salazar et al. [128] obtained very similar results in the three models, when calculating the coefficients for the three cases, in the first model which predicted the internal temperature they obtained a value of 0.976, while the model that predicted the internal humidity obtained a value of 0.982 and the third model for the internal temperature and internal humidity obtained values of 0.975. Salazar et al. results show that the third model is less efficient in predicting the output variables, however it is not significant and has the advantage of predicting the output variables at the same time.

One way in which you can "feedback" to a FFNN is through delays in the input variables, this process consists of considering certain outputs also as inputs; these outputs, when considered inputs, have a certain delay to be able to feed the network with new information on the variable of interest. Alipour et al. [129] tested three network configurations with this type of delay to forecast the relative humidity, the infrared index, the light index and the internal temperature of the greenhouse. For example, to predict internal temperature, Alipour et al. [129] found that the optimal structure should use a direct feeding neural network with 7 input delays and 5 neurons in the

hidden layer. In this way, a way to build FFNNs that have been exploited very little is presented, and of the studies analyzed only Alipour et al. [129] and Outanoute et al. [130] explore this path.

*6.5. The Hidden Layer of ANNs and Their Importance in Prediction of Greenhouse Microclimate*

In the documents presented, Taki et al. [124,131] emphasize the importance of the hidden layer in prediction of the greenhouse microclimate, that is, the number of optimal layers and neurons. Although the method for determining the number of neurons and layers is more of a trial and error process, Taki et al. [124] mention three circumstances to consider when building a neural network for microclimate in greenhouses: the first is that the performance of the network can improve as the number of hidden neurons increases; the second is that too many neurons in the hidden layer can cause overfitting problems, which influences learning and memory of data, but impairs the ability to generalize; and finally the third, if the number of neurons is too low it is possible that the neural network loses the ability to learn. Taki et al. [131] through an RBF network determined that the optimal hidden layer is built by three layers with 21, 9 and 9 neurons, respectively, for the prediction of internal air temperature, plant temperature, and greenhouse soil temperature. They indicate that adding more neurons than those established in your specific case does not significantly increase the predictive power. Furthermore, in the case of the RBF network, the process of increasing the coefficient $R^2$ also depends on the values in the propagation parameters.

*6.6. Learning Algorithms in the ANNs*

Training algorithms have a great influence on the efficiency of NN sand choosing which the best will depend on various factors. Outanoute et al. [130] tested three training algorithms, the momentum gradient descent (GDM), the quasi-Newton BP Broyden–Fletcher–Golfarb–Shano (BFGS), and the resilient BP algorithm (RProp). For each network, the number of nodes in the hidden layer was also varied, since the efficiency of the network varies for each case and the number of optimal neurons was searched for each network. The results in the training stage showed that the BFGS network has better performance, the mean square errors (MSE) were for the internal temperature of 0.0022 and for the internal humidity of 0.0034, while for the GDM they were 0.1877 and 0.1143 and, for the RProp they were 0.0349 and 0.0433. Ferreira et al. [112], when testing online or offline methods, determined that the LM algorithm has an advantage over techniques such as resource allocating network (RAN), orthogonal least squares (OLS) algorithms, among others. The smallest root means square error (RMSE) off-line was 0.0108 with a network of 8 neurons in the hidden layer, and an online network obtained a value of 0.0072 with a similar structure. That is, better results were achieved either online or offline with the LM methods compared to other hybrid and adaptive.

*6.7. Database for ANNs and Prediction of Greenhouse Microclimate*

The database is highly relevant for the proper functioning of the neural network. Throughout this document, different works have been presented where the ANNs are applied for the prediction of the greenhouse microclimate, and it can be seen that the optimization of the network is linked to its type and structure. However, the elaboration and collation of data takes equal importance, since the effectiveness of training and building the network depends on it. At least three stages can be mentioned in which the database becomes the pillar of a neural network: training, validation and the testing phase. These stages, in the same way, can be considered as the stages through which a neural network must pass.

The amount of data that a black box model requires specifically if we are talking about an ANN, must be from a relatively large sample. Delimiting the optimal amount is a process of trial and error, as is the choice of hidden layers and the number of neurons. Having a sample with a really large data set benefits the prediction power, however, the training and prediction process of the network will be affected by less-efficient processes. Applying techniques such as proposed by Seginer et al. [125] and He et al. [60] can simplify excessively large databases and optimize training processes of

the NNs. Taki et al. [124] mention that a 12-month database is ideal for an ANN that predicts the greenhouse microclimate, although the vast majority of the studies presented do not consider this period. Dariouchy et al. [123] only consider a database of 29 days; in the training phase it used most of the data (22 days) and for the test phase, used the rest (7 days). Outanoute et al. [130] only considered a three-day period, although the amount of collated data consisted of 25,750 values, which divided them into 70% for network training, 15% for validation and 15% for the phase test. Alipour et al. [129], also used a three-day database, the difference being these days were not consecutive and it was at a specific time (from 10 a.m. to 8 p.m.). Although the period of time is important, the amount of information generated with that data and the management of it during those periods is more relevant. Salazar et al. [128] used a base of 14,490 data, 50% used them for training, 25% for verification and 25% for tests. Database management can be seen in Laribi et al. [126], since they used different databases for the different processes, in the training they used a one-day data set with temperature ranges from 6 °C to 14 °C; for the tests two days of different years were used, but with temperature ranges from 6 °C to 9 °C. In other words, a model can be built with a database that already has certain specific information for the network.

Being able to delimit the minimum data required in a network is an empirical process, as well as choosing the amount used in training, verification and testing. Cases such as those of Seginer et al. [125] used a registry of 3076 data or as those of Taki et al. [124] that its compiled database represents a set of values of one day. The team is currently developing a neural network using 70,032 data collected during 1 year, of which 50% will be used for network training, 25% for verification and 25% for testing. However, having a large data set can also impair the operation of the network, since the presence of trivialities is more likely. For this reason, it is necessary to use techniques that clean the database when this type of problem occurs, as well as before constructing the network, delimit the most important parameters for the study variables, seeing the influence of the variables of input with the output variables using techniques such as sensitivity analysis.

## 6.8. Artificial Intelligence

Artificial Intelligence (AI) researches and builds intelligent software and machines, provides a particular solution to a particular defined complex problem, is made up of branches such as genetic algorithms, particle swarm optimization, simulation and ANNs and hybrid models (two or more of the above) [206,207]. AI consists of mapping non-linear behavior between inputs and outputs of processes [208]. AI consists of a large number of practical tools that allow solving difficult problems tasks that require biological or human intelligence, with functions such as perceptron, recognition, Decision-making and control combines brain science and related fields, such as cognitive science and psychology [209]. The AI allows the prediction of thermal properties of biomass, tools such as ANN have proven to be vital in the development of research in the prediction of biomass energy, which in turn could be used in the control of greenhouse microclimates. NNs are flexible to accommodate non-linear and non-physical data; however, they require a large multidimensional data set to reduce the risk of extrapolation. [210]. AI employs quite different mathematical and algorithmic approaches, from operational research restricted programming, DL and ML [211].

DL expands on classic ML by adding more depth to modeling. Its advantage is feature learning that is, automatic extraction of features from raw data and quick resolution of complex problems. The DL is made up of various components, such as convolutions, grouping layers, fully connected layers, gates, memory cells, activation functions, and encoding/decoding schemes, depending on the network architecture used, such as the aforementioned convolutional neural networks (CNN) [212].

## 6.9. Future of Deep Learning in Greenhouse Agriculture

DL has demonstrated a great capacity in pattern recognition and ML. One of the main tasks of this type of network is to learn to actively perceive patterns by sequentially directing attention to relevant parts of the available data [213]. The advantage of DL over conventional networks is the possibility of developing simulated data set to train the model, which would allow solving real-world problems, such as greenhouse systems. In [214] the various applications of DL in

agriculture are exposed, however, its use in greenhouses is still lacking. Of the studies that have been carried out of DL in greenhouses we can find the one carried out in [212], who propose a new deep RNN, with a long short-term memory neural network (LSTMNN) model to predict the stem diameter, or tomato performance problems using environmental parameters such as $CO_2$, humidity, radiation, outside temperature and indoor temperature. One of the main disadvantages of this method is exposed: the large amount of data necessary for the training process.

The BP network is the basis of the vast majority of DL algorithms, it also allows models that consist of multiple layers of processing to represent data with multiple levels of abstraction, so its application in agriculture has begun to be studied. In the field of smart farming (SF) it can be used for the detection of plant diseases, weed control, and plant counting through image recognition. CNN, RNNs, and generative adversarial networks (GANs) being the most viable types of deep networks in this field [215].

CNN models are an extension of the DL. They consist of MLP networks that involve multiple pools and fully connected layers, learn and optimize filters on each layer through the back-propagation mechanism. These trained and learned filters extract features that distinctively represent the input image. This type of model has managed to overcome state-of-the-art algorithms and since then has become the most advanced method in many data processing tasks. Currently, CNN architectures are trained from scratch or adjusting pre-trained architectures. Using pre-trained architectures allow transfer learning to be used. Transfer learning consists of using the learning of models that have been previously trained with large data sets from other systems, in other problems or similar systems [216].

CNN has a great capacity in image processing, which makes it widely used in agricultural research. The challenge with the use of information is to interpret the collected images. Interpreting satellite images using CNN and GA has become a useful decision-making strategy, especially for PA [215]. Furthermore, they can also be used in weather forecasting, which is key for agriculture [217].

### 6.10. Future of Hybrid ANNs in Greenhouse Agriculture

The use of the combination of ANNs with mathematical models has been little explored, however, as can be seen in Yousefi et al. [218] and Linker et al. [205], the approach can be considered from two perspectives: First, using techniques such as fuzzy logic for optimization in the random choice of the initial parameters and second, to use the physical models for the generation of synthetic data that help the network in the learning process, minimizing the errors due to the lack of information that a base can present of data in situ. ANN hybrid models have the potential to provide forecasts that work well compared to more traditional modeling, such as the use of ANN models optimized by PSO and GA that have shown good prognostic results of energy requirements [219].

### 7. Guidelines for the Application of Neural Networks in Greenhouses

In this review we have presented the application of ANNs in the prediction of the microclimate in greenhouses, their use in energy optimization, as well as other applications in greenhouses. Of these topics addressed, it should be noted that the potential of the ANNs continues to be promising for future research. Although studies have presented ways in which one should delve further, such as the use of physical models in conjunction with NNs, the work is still scarce.

Physical models for the creation of synthetic data is a good strategy to feedNNs, since it would complement in situ data and would allow confronting the possible problems that the nature of this type of data entails. Linker and Seginer [205] presents two possible configurations, in series and in parallel. However, a serial configuration has not been tested and its use in conjunction with statistical tools such as PCA would optimize the data selection process for a neural network. Also, the structure of hybrid networks could be expanded and the serial and/or parallel configurations could be used in conjunction with optimization algorithms such as GA, PSO, among others.

The use of WSN in greenhouses is part of SA, offering advantages not only in data collation and greenhouse climate control, but also in the energy consumption of this type of device. The application of the ANNs in the WSN of the greenhouses would also be an interesting topic to

develop, since on the one hand there is the automatic reprogramming of the wireless sensors and on the other the forecast of the greenhouse variables collated using this sensor network.

RNNs are little used compared to FFNNs, but their use in greenhouses in conjunction with image analysis can be of help in the identification of diseases and pests in greenhouses. Also, DL and CNNs are tools that would facilitate these tasks. Likewise, the forecast of the microclimate in greenhouses using analysis of thermographic images and CNNs would be something interesting to apply.

CNNs in the study of growth and transpiration crops is an issue that would be worth developing. In traditional methods, these processes represent a mathematical challenge while CNNs would simplify them with help of morphological and thermal analysis.

The application of new technologies such as 4.0 in greenhouses opens the panorama of carrying out work with a perspective on integration and exchange information. The studies presented have been developed with data obtained from a single greenhouse; that is, the object of study has been the case of a particular greenhouse. However, with the IoT, the WSN, the CC, among others, information from various points (greenhouses) could be accessed at any time. ANNs can be developed that use this data, but first it would be necessary to make a reliable database. Although each greenhouse is in the same region or has a similar outside climate, variations would still be present. However, synthetic data could be used to minimize these variations. The WSN could be trained to detect diseases and plagues in crops and this information can be used in nearby greenhouses to predict the presence of these afflictions and take the necessary preventive measures.

## 8. Conclusions

This review presented different studies of ANNs in greenhouses. Most of the studies had a focus on the prediction of the microclimate where the use of feedforward networks is the most used architecture. However, RNNs are less used and it is necessary to explore different architectures and training methods in order to determine the advantages and disadvantages they may have compared to feedforward. Likewise, the development of this type of network will allow the use of new methods such as DL in tasks that facilitate production under greenhouses.

Network training is one of the processes where optimization techniques must be measured in order to reduce calculation times and data management. The use of statistical tools such as PCA is viable, however, the application of methods such as GA, FL, and PSO should be considered in the same way in more complex architectures such as RNNs and not only in feedforward networks.

Unlike the physical models, ANNs take just a few minutes to finish an indoor climate forecast, considering that many unknown factors are involved and are not possible to study with physical models. ANN and physical models' combination would allow a better prediction of a microclimate, however, this hybrid network's construction is poorly investigated and hence this network should be studied. Within other applications of NNs in greenhouses, the evaluation and prediction of plagues and diseases in crops can be driven by technologies such as image analysis in combination with DL models such as CNNs. Similarly, microclimate prediction might be feasible with these techniques since a greater amount of information can be handled by these methods compared to traditional ANN models.

An important guideline for future works is the integration and exchange of information using 4.0 technologies. The role of the ANNs is to develop predictive models that take advantage of the information generated and its management.

**Author Contributions:** A.E.-G. writing—original draft preparation; G.M.S.-Z. advised on planning and manuscript writing; M.T.-A. collaborated in topics structuring of neural network topics, E.R.-A. planned agriculture section 4.0, A.G.-B. provided, organized and analyzed application of artificial neural networks section. All authors have read and agreed to the published version of the manuscript.

**Funding:** This research received no external funding.

**Acknowledgments:** The authors would like to thank CONACyT and UAQ for their financial support. Also, thanks to Mariana Montserrat Flores Nieves, PhD student in Biosystems Engineering, for the support in the realization of the figures.

**Conflicts of Interest:** The authors declare no conflict of interest.

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
