# Peer review of "Applications of Artificial Neural Networks in Greenhouse Technology and Overview for Smart Agriculture Development"

_applsci, doi:10.3390/app10113835_

Round 1

Reviewer 1 Report

Dear Authors,

thank you by your answers.

The text improved, but it is not necessary to write "as mentioned...". I recommend to rewrite paragraphs with such term and refer to the proper section if it is really necessary.

I understood the explanation about ANN and it not necessary to write that it is difficult to separate training set and test set. It is something empirical and there is several approaches about how to deal with such question that just split in 50% 25% 25%. But such data organization is a valid option.

Author Response

Q1. The text improved, but it is not necessary to write "as mentioned...". I recommend to rewrite paragraphs with such term and refer to the proper section if it is really necessary.

R. We acknowledge this comment. Fixed and removed the parts where "as mentioned" was written. The recommendation was followed and the modified lines were:
125, 263, 989, 1058, 1139, 1160.

Q2. I understood the explanation about ANN and it not necessary to write that it is difficult to separate training set and test set. It is something empirical and there is several approaches about how to deal with such question that just split in 50% 25% 25%. But such data organization is a valid option.

R. We acknowledge this comment. It is indeed an empirical process, one of the purposes of the fragments included in lines 1146-1159 and 1161-1171 is to explain such a process. However, the annotation was made according to what was commented to make the idea clearer.

Reviewer 2 Report

This paper presents different studies of ANNs in greenhouses. Most of the work discussed is based on the prediction model. This is relevant from the angle of considering the microclimate, while exploring the different architectures and training methods. The paper deals with setting up a contrast between traditional and new ANNs models. Few important considerations to be made in this paper are – Restructuring of the paper, generalization and categorization of the discussed approaches are missing and overall from reader perspective paper has missing dots and difficult to co-relate (for instance, use diagrams to group the approach and for better understanding).

Few inline remarks are given below -

[Line61] how solar energy represents a viable option as an energy source instead of the traditional ones? It needs reference and a comparison with other evolving energy efficient techniques.

[Line88] be consistent while using comma, while referring each section.

[Line85] it would be nice to give a short explanation for 4.0 technologies and map it’s requirement/demand. Currently, it is unclear which technologies are applied in smart agriculture.

[Line138] Figures 1, 2 are important and referring them first time inside brackets is inappropriate. Also, Fig 2 needs explanation.

[Line169] why Gaussian is capitalized and hyperbolic not? Try to be consistent throughout the paper.

[Line170] table is inside brackets, it needs explanation. Also, it would be nice to list explicit key difference making properties for each function in the table.

[Line204]” in this case”, which case you are referring to? Bring modularity in sections.

[Line218] Be consistent throughout the paper, (Neurons or neurons)

[Line219] In feedforward neural network, explain the problem you mentioned or give the citation. Backpropogation training also needs a reference.

[Line272] Hebbian learning is introduced but not explained or referred.

[Overall] All the tables and Figures are referred inside brackets, which is distracting the reader. It is advised to explain and refer like “as shown in Fig. 3”

[Line 333] Reference for supervised and unsupervised learning? Also, need to add SOTA in same context.

[Line 352] Single sentence is of four lines which can be easily broken down for better understanding. Also, references are not properly cited.

[Line 365] which harvest you are talking about? Also, what do you mean by “this due to the aforementioned”. Greenhouse microclimate, needs the details for the state of the art. Recent work is not included.

[Table 2] In two of the references, training method field is missing.

[Section 4] Energy optimization of greenhouses, discussion of control elements is missing. Also, other energy harvesting methods that uses prediction method.

[Section 5] Other applications may include the prediction model used in sensor networks as well.

[Line 668] Needs a reference for marked several issues, models and generalization of models.

[Section 6.5] It needs generalization of the database used in the experiments.

[Section 6] Use of DL and ML in sensor network for prediction and comparing with your section on energy consumption can be interesting addon.

[Section 6.9.1] Prescision agriculture and Internet of Things and Smart agriculture comes just before conclusion. However, this can be covered initially before discussing different approaches.

Author Response

Q1. This paper presents different studies of ANNs in greenhouses. Most of the work discussed is based on the prediction model. This is relevant from the angle of considering the microclimate, while exploring the different architectures and training methods. The paper deals with setting up a contrast between traditional and new ANNs models. Few important considerations to be made in this paper are – Restructuring of the paper, generalization and categorization of the discussed approaches are missing and overall from reader perspective paper has missing dots and difficult to co-relate (for instance, use diagrams to group the approach and for better understanding).

R. We acknowledge this comment. Upon attending the indications made, the article was restructured. In addition, two diagrams were added. The first one to facilitate the understanding of the two approaches under which modeling is carried out in greenhouses, and the second that summarizes the type of neural network, the architecture and the training method most used in greenhouses. The diagrams are presented below (lines 103 and 943):

Diagram 1: Greenhouses and approach to describe them

Diagram 2: Artificial neural networks on greenhouse microclimate prediction

Q2. [Line61] How solar energy represents a viable option as an energy source instead of the traditional ones? It needs reference and a comparison with other evolving energy efficient techniques.

R. We acknowledge this comment. We support this information and the paragraph included by lines 60-74 was modified as follows:

In addition to the microclimate and its control, another topic of interest derived from the production of greenhouse crops is its energetic consumption and how solar energy represents a viable option as an energy source instead of the traditional ones (fuel and electricity). Solar energy is better than traditional sources, because fuels are no renewable and represent high cost [39]. Traditional energy sources can be replaced with other sustainable energies, such as solar energy, wind energy [40], biomass [41,42,43], geothermal energy [44,45,46], cogeneration systems [47,48], among others. However, use of solar photovoltaic cells or solar thermal energy in greenhouses are more widely used and can commonly be combined with other sustainable energy systems [49]. Solar greenhouses provide a controlled system cultivation, the most focus is to reduce heating energy requirements, i.e. the heating requirement is largely derived from the sun [39]. Furthermore, how solar energy represents a primary element in the heating of greenhouses and makes it possible to minimize production costs [50]. Several studies have been carried out in which energy savings are sought, where methods such as genetic algorithms (GA) have been applied to optimize energy collection [5140], also physical models [52–54], as well as computational fluid dynamics (CFD) techniques to predict the microclimate of solar greenhouses [55–57].

Q3. [Line88] be consistent while using comma, while referring each section.

R. We acknowledge this comment. The correction was made as follows (lines 96-101):

The structure of this document is: Section 2, gives an ANNs explanation, different activation functions, types and different learnings of ANNs. Section 3 presents the neural networks application for the prediction of microclimates in greenhouses. Section 4 shows neural network applications in greenhouse energy optimization. Section 5, indicated other studies and ANNs in greenhouses applications. Finally, section 6 addresses the challenges in the development of neural networks in greenhouse agriculture.

Q4. [Line85] it would be nice to give a short explanation for 4.0 technologies and map it’s requirement/demand. Currently, it is unclear which technologies are applied in smart agriculture.

R. We acknowledge this comment. There is a topic dedicated to the explanation of agriculture 4.0 (lines 779-799). In addition, sub-topic 6.1.2 (lines 848-878) details the technologies involved in smart agriculture

Q5. [Line138] Figures 1, 2 are important and referring them first time inside brackets is inappropriate. Also, Fig 2 needs explanation.

R. We acknowledge this comment. Corrected and gives a brief explanation of the figures. In figure 2 (figure 3 after correction) each of the parts that make it up was detailed as well as the formulas integrated in it. The explanation was as follows (lines 172-176):

Figure 3 presents the structure of a simple neuron. The processing of the information in a neuron begins with the inputs Xn, they are weighted and added up before going through some activation function to generate its output, this process is represented as ξ= ΣXi·Wi. For each of the outgoing connections, this activation value is multiplied by the specific weight Wn and transferred to the next node. If consider a linear activation, the output would be given by y=α(wx+b) [65].

Q6. [Line169] why Gaussian is capitalized and hyperbolic not? Try to be consistent throughout the paper.

R. We acknowledge this comment. Corrections were made

Q7. [Line170] table is inside brackets, it needs explanation. Also, it would be nice to list explicit key difference making properties for each function in the table.

R. We acknowledge this comment. A brief explanation of the table was made and more information was added to attend to the indication, however due to lack of space and format, we opted to add it in the text (lines 210-233):

Table 1 shows the activation functions commonly used in neural networks. The behavior of neurons is defined by these functions. If the transfer function is linear and the network is multi-layered, it can be represented as a single-layer network, since it is the product of the weight matrices of each layer and will only produce positive numbers over the entire range of real numbers. On the other hand, non-linear transfer functions (such as the sigmoid function) between layers allow multiple layers to provide new capabilities, adjusting the weights to obtain a minimum error in each set of connections between layers [68,69]. Linear functions are generally used in the input and output layers, while nonlinear activation functions can be used for the hidden and output layers [70].

The most used nonlinear activation functions are the sigmoid and hyperbolic tangent. Hyperbolic and sigmoid tangents are mainly used because they are differentiable and make them compatible with the back-propagation algorithm. Both activation functions have an "S" curve, while their output range is different [71]. The sigmoid function is the most used activation function in ANNs. This function varies from 0 to +1, although the activation function is sometimes sought to oscillate between –1 and +1, in which case the activation function assumes an antisymmetric form with respect to the origin, defining it as the hyperbolic tangent function [72 -74].

Direct implementation of the sigmoid and hyperbolic tangent functions in hardware is impractical due to its exponential nature. There are several different approaches to the hardware approximation of activation functions, such as the piecewise linear approximation. Linear part approximations are slow, but they are the most common way of implementing activation functions [75]. In addition, it uses a series of linear segments to approximate the trigger function. The number and location of these segments are chosen so that errors and processing time are minimized [76].

The Gaussian activation function can be used when finer control over the activation range is needed [69]. Furthermore, it can uniformly perform continuous function approximations of various variables [77].

Q8. [Line204]” in this case”, which case you are referring to? Bring modularity in sections.

R. We acknowledge this comment. Cases in which the same situation occurs were corrected and attended to.

Q9. [Line218] Be consistent throughout the paper, (Neurons or neurons)

R. We acknowledge this comment. A format for the entire paper was run and handled

Q10. [Line219] In feedforward neural network, explain the problem you mentioned or give the citation. Backpropogation training also needs a reference.

R. We acknowledge this comment. We support this information and it was corrected throughout that topic.

Q11. [Line272] Hebbian learning is introduced but not explained or referred.

R. We acknowledge this comment. A brief explanation was made as follows (lines 326-328):

Hebbian learning involves synapses between neurons and their strengthening when neurons on both sides of the synapse (input and output) have highly correlated outputs [87] as shown in Figure 5b.

Q12. [Overall] All the tables and Figures are referred inside brackets, which is distracting the reader. It is advised to explain and refer like “as shown in Fig. 3”

R. We acknowledge this comment. The correction was made and the suggested indication followed as well as a brief explanation of the figures when required.

Q13. [Line 333] Reference for supervised and unsupervised learning? Also, need to add SOTA in same context.

R. We acknowledge this comment. We support supervised and unsupervised learning and information about SOTA was added as follows (lines 394-404).:

The exposed learning techniques have allowed the development of more advanced algorithms such as SOM (Self-Organizing Maps) and SOTA (Self-Organizing Tree Algorithm), which are time series clustering algorithms based on unsupervised neural networks [93]. The SOM is a known data analysis tool for tasks like data visualization and clustering. One of the disadvantages of this tool is that the user must select the map size. This may lead to many experiments with different sized maps, trying to obtain the optimal result. Training and using these large maps may be quite slow [94]. While the SOTA permits to classify in the initial levels the groups of patterns that are more separated from each other and to classify patterns in the final layers in a more accurate way [95]. These techniques open up the possibility of not only learning connection weights from examples, but also learning a neural network structure from examples. This thanks to the fact that a neural network can be built automatically from the training data by SOTA methods [96].

Q14. [Line 352] Single sentence is of four lines which can be easily broken down for better understanding. Also, references are not properly cited.

R. We acknowledge this comment. The topic was restructured and the suggestion was followed to facilitate understanding (lines 418-478).

Q15. [Line 365] which harvest you are talking about? Also, what do you mean by “this due to the aforementioned”. Greenhouse microclimate, needs the details for the state of the art. Recent work is not included.

R. We acknowledge this comment. The indicated was corrected and a brief state of the art of the works that fit the topic of the review was added. In addition, the issue was restructured as shown below (lines 418-478):

The greenhouses are complex and non-linear systems, likewise, it is a means to achieve a controlled agricultural production [7]. Greenhouse production systems have a complex dynamic impulsed by external factors (meteorological), control mechanisms (ventilation openings, exhaust fans, heaters, evaporative cooling systems, etc.) and internal factors (crops and internal components) [6].

 Concerning the microclimate of the greenhouse and its control, the crop represents the central element, but also the most complex of the system. Due to this complexity and the great diversity of crops in greenhouses, it is common to consider only certain general issues that are more relevant to the response of crops in relation to greenhouse microclimate [99].

The climate in greenhouses refers to the set of environmental variables in this system that affect the growth of crops and their development [100]. Greenhouse microclimate control has received considerable attention in recent years due to its great contribution to the improvement of crop yield [29, 101, 13]. The different factors such as temperature [102, 103], relative humidity [104], amount of CO2 [105, 106] are analyzed to predict different events implementing artificial intelligence, statistics and engineering [107,49, 108, 109, 110].

Numerous greenhouses use a conventional control, but this control strategy may not be suitable to guarantee the desired performance [15,26]. In this scenario, various strategies and control techniques have been proposed like generalized predictive control [111], optimal control [112], model predictive control [16, 113], neural networks control [114], fuzzy control [115,116,117], robust control [15,118] and linear-quadratic adaptative control [119]. The vast majority of these proposals are simulations of the behavior of the variables and possible control against these changes and are also focused on a specific crop [120].

The application of neural networks in the control of microclimates is a topic that has currently gained interest, NN provides reliable models that can reflect the non-linear characteristics of the greenhouse that are difficult to solve using traditional techniques, and does not require any prior knowledge of the system and is very suitable for modeling dynamic systems in real time [121, 122].

Temperature and humidity are of the most relevant parameters in the greenhouse microclimate, since they have complex exchanges and interactions of heat and mass between the inner air, other elements of the greenhouse and the outside. Building a model is a difficult assignment with simple mathematical formulas or transformation functions. However, the method of building models with ANNs has a great capacity for mapping nonlinear functions, which is applied to many production process systems [123].

For the network design, air temperature and humidity of the greenhouse air are generally considered as outputs, this due to the aforementioned. However, setting the inputs is more complicated and requires an understanding of the system. It is not convenient to consider a large number of inputs, as this could cause uncontrolled extrapolations instead of increasing the estimation power. Three elements can be considered in order to consider an input variable: 1) correlation of selected input with other inputs, 2) physical dependence nature of the output and the input 3) input variable range. Third point, solar radiation at greenhouse temperature can be considered as an example [98].

Q16. [Table 2] In two of the references, training method field is missing.

R. We acknowledge this comment. The information was completed, in one of the references it was not defined which training method was used as in the others.

Q17. [Section 4] Energy optimization of greenhouses, discussion of control elements is missing. Also, other energy harvesting methods that uses prediction method.

R. We acknowledge this comment. The information was completed with the elements indicated as shown below (lines 640-698):

Another problem of interest in greenhouses is the optimization of energy consumption derived mainly from heating and, ventilation systems, among other control elements [145]. Optimal control strategies, for the most part, are based on mathematical models for calculating greenhouse energy consumption and mathematical methods to minimize total energy consumption. An example is the use of steady state energy balance models. The use of this type of model is not new [146], nor is its use for real-time energy optimization [9]. However, the implementation of these techniques with sustainable technologies such as photovoltaic (PV) collectors have allowed predicting performance and establishing better systems for energy consumption [147]. Furthermore, greenhouses with systems that optimize energy consumption must assess heating needs before being implemented, and one of the ways to do this is through mass flow and energy transfer models [148]. Currently these models are still being developed to predict heating requirements. They make it possible to resolve different issues related to this topic, such as the forecast of the hourly energy requirements based on the entry of the parameters of environmental control inside the greenhouse, the physical and thermal properties of the crops and the construction materials [149]. In addition, CFD-based energy saving and system performance models have been proposed [150].

The use of other types of models such as based optimization techniques such as PSO and GA have also shown good results [151], as well as with neural networks [10].

Energy consumption is largely derived from two factors that influence the aforementioned control elements, temperature, and humidity. Trejo-Perea et al. [152] developed a predictor of energy consumption for greenhouses from an MLP, also compared the ANN model with a non-linear regression model. The results obtained show that the prediction power of the network is superior to the regression model with an accuracy significant level (95%). Regarding the structure of the network, a cascade architecture was carried out where the input variables were temperature, relative humidity, time and electrical consumption, on the other hand, the output variable considered was the electrical consumption. Several MLP models were tested, where the hidden layer was the only variant with five, four, three and two neurons. While the Levenberg-Marquardt reverse propagation algorithm was used for the learning procedure. The MLP model with the best results was the model with three nodes in the hidden layer, also compared to the regression model.

The use of elements that help the energetic production in greenhouses is also a topic of interest, in the same way, its energy management and optimization. Photovoltaic modules are a viable option for this task, Pérez-Alonso et al. [153] developed a photovoltaic greenhouse, where the use of ANNs focused on the prediction of instantaneous production of the system. The network used was a feedforward trained using an LM algorithm. The input variables considered were ambient temperature, relative humidity, wind speed, wind direction, and radiation. As output variables, only photovoltaic energy production was considered. The hidden layer of the network consisted of 140 neurons, the tests were obtained in 1 second and prediction errors for the instantaneous production of electricity below 20 Watts.

Other studies have used ANNs to predict greenhouse production using the amount of energy use as a basis. Such is the case of Taki et al. [154] who, through an MLP network, predicted greenhouse tomato production. They used as inputs the energy equivalences of chemical products, human energy, machinery, chemical fertilizers, diesel fuel, electricity, and irrigation water. The architecture used consisted of 7 inputs, 10 neurons for each of the two hidden layers and one output (tomato production). No transfer function was used for the input layer, for the hidden layers a hyperbolic tangent transfer function was used, and for the output layer a linear transfer function was chosen. The results revealed that diesel fuel (40%), chemical fertilizers (30%), electricity (12%) and human energy (10%) consumed most of the energy. The comparison between the ANN model and the multiple linear regression model (MLR) showed that the ANN model predicts the output performance significantly better than the multiple MLR model.

The development of new control strategies influence energy costs by reducing the energy consumption of greenhouses. However, the potential for energy saving control seems to be over-estimated. Climate control strategies for energy saving have been developed [155], from the analysis of greenhouse roofing materials and how these affect energy consumption [156] to the use of thermal screens and how they can reduce consumption of energy at night [157]. Likewise, the response of the crop has been investigated when applying techniques for energy saving [158], however, it is necessary to explore more methods beyond those exposed and the application of neural networks as a viable tool.

Q18. [Section 5] Other applications may include the prediction model used in sensor networks as well.

R. We acknowledge this comment. Added a part that discusses the suggested topic as shown below (lines 758-777)

Wireless sensor networks (WSN) are a new form of distributed computing and are encompassing a wide variety of applications that can be implemented with them [165]. In greenhouses it is primarily concerned with collecting environmental information and sending it to the grouping nodes via wireless data link. WSN is a type of self-organizing wireless network that takes data at its core [166]. The role of this technology and ANNs is that they are a good combination for controlling greenhouses. WSN can be used to monitor CO2 concentration [167]. Zhang et al. [168] carried out a greenhouse control system using a WSN to collect data on temperature, humidity and CO2 concentration. They related the internal environmental factors and the actuators of the system for the implementation of a fuzzy rule and combined with a neural network. The fuzzy neural network consisted of three inputs and six outputs to improve control precision. Also Ting et al. [169] measured and collected real-time data on air temperature, humidity, CO2 concentration, soil temperature, soil moisture, and light intensity using WSN. The measurement of these parameters was to predict the photosynthetic rate of plants and in turn to quantitatively regulate CO2. The prediction model was established based on a BP neural network. The environmental parameters were used as input neurons after being processed by PCA, and the photosynthetic rate was taken as the output neuron.

There are many important areas where WSN can improve. One of the aspects to consider is to give the sensor networks the ability to reprogram themselves wirelessly, allowing users not to physically interact with the sensor nodes. This wireless reprogramming can be based on the concept of neural networks as proposed by Cañete et al [165], and thus be able to implement it in greenhouses.

Q19. [Line 668] Needs a reference for marked several issues, models and generalization of models.

R. We acknowledge this comment. The information was completed with references that frame the topics covered (lines 960-968).

Q20. [Section 6.9.1] Precision agriculture and Internet of Things and Smart agriculture comes just before conclusion. However, this can be covered initially before discussing different approaches.

R. We acknowledge this comment. The article was restructured so that the topics of Precision agriculture, Internet of Things and Smart agriculture are dealt with before the discussion of the perspectives of neural networks and their future in greenhouses. Being the subtopics 6.1, 6.1.1 and 6.1.2, respectively.

Reviewer 3 Report

The authors described important and challenging issues related to the problems of greenhouse technology. They presented it in a form of an overview in which they described advanced technological solutions related to the examined area. The comparative analysis is dominated by emphasizing possibilities of various artificial neural network approaches. That part is the strong point of the paper.

It is rather trivial to describe a structure and topology of a general ANN (Chapter 2), such information could be found in many textbooks. More important parts of the paper deal with implementation of various topologies for greenhouse analyses, especially representation of greenhouse microclimat parameters and, above all, comprehensive lists of input variables confronted with output variables. Chapter 3 is definetely a valuable part of the paper.

The authors also mentioned other promising technologies like IoT, ML, image analysis and big data (Chapter 6, 6.6. 6.7), but that aspect was treated too superficially to keep it in the paper, I would advise to concentrate on facts, and either to explore those modern technologies with respect to greenhouses or to remove it.

There are some controversies about the comparison of  classical models versus ANN (Chapter 6.1 Classic models VS ANNs). I do not criticize the text, I simply discover its incompleteness. Classical structural modeling methods, with the support of inverse modeling to estimate (identify) model coefficients are still in the game as the only models enabling us to understand and predict the complex greenhouse processes.

Chapter 6.9 Agriculture 4.0 and the ANNs should be published as a separate paper, after major supplementing based on the current literature.

As the conclusion, the paper is a significant contribution to the field of climate control in greenhouses with the use of advanced artificial neural networks. Other discussions concerning so called classical methods, and modern AI approaches, should be either enhanced or removed.

Author Response

Q1. It is rather trivial to describe a structure and topology of a general ANN (Chapter 2), such information could be found in many textbooks. More important parts of the paper deal with implementation of various topologies for greenhouse analyses, especially representation of greenhouse microclimate parameters and, above all, comprehensive lists of input variables confronted with output variables. Chapter 3 is definitely a valuable part of the paper.

R. We acknowledge this comment. Chapter 2 was added to the article because previously another reviewer requested that section be added.

Q2. The authors also mentioned other promising technologies like IoT, ML, image analysis and big data (Chapter 6, 6.6. 6.7), but that aspect was treated too superficially to keep it in the paper, I would advise to concentrate on facts, and either to explore those modern technologies with respect to greenhouses or to remove it.

R. We acknowledge this comment. That is why section 6 was restructured, the subtopics that now comprise this section are:

6.1 Agriculture 4.0 and the ANNs

6.1.1 Precision agriculture and Internet of Things

6.1.2 Smart agriculture

6.2 Artificial neural networks and greenhouses

6.3 Classic models VS ANNs

6.4 The input variables in the ANNs and in the prediction of greenhouse microclimate

6.5 The hidden layer of ANNs and their importance in prediction of greenhouse microclimate

6.6 Learning algorithms in the ANNs

 6.7 Database for ANNs and prediction of greenhouse microclimate

6.8 Artificial Intelligence

6.9 Future of deep learning in greenhouse agriculture

6.10 Future of hybrid ANNs in greenhouse agriculture

Q3. There are some controversies about the comparison of classical models versus ANN (Chapter 6.1 Classic models VS ANNs). I do not criticize the text; I simply discover its incompleteness. Classical structural modeling methods, with the support of inverse modeling to estimate (identify) model coefficients are still in the game as the only models enabling us to understand and predict the complex greenhouse processes.

R. We acknowledge this comment. In the topic "Classic models VS ANNs" the information is supplemented addressing the advantages, importance of the classic models and at what times they have been used in greenhouses, as shown below (lines 989-1004):

Greenhouse climate models can be classified into two categories [127]: models for the design of new greenhouses and models for climate control of existing structures. The latter are also known as classic models, they are based on steady state energy balances. The number of parameters in this type of model is small compared to MIMO (multiple input multiple output) black box models, this being one of its main advantages.

On the other hand, mechanistic models provide a clear physical explanation of the greenhouse environment such as static [200,201] and dynamic models [202]. Static models are based on the static energy balance of the greenhouse components and usually their heat storage capacities are not considered [203]. The relevance of physics-based models in greenhouses is that they take the physical parameters that describe the system, they can include the location of the greenhouse, local weather conditions, geometry, construction materials, hours of operation, systems of air conditioning and settings. That is, they allow its use for the design phase and help to evaluate the energy performance of the greenhouse [204,127]. However, the current state of climate control still has much room for improvement [205] and optimal control of greenhouse environments can be improved by combined models to allow selection of greenhouse designs and control algorithms to maximize the room for improvement. benefit such as models based multi-objective optimization [206,207].

Q4. Chapter 6.9 Agriculture 4.0 and the ANNs should be published as a separate paper, after major supplementing based on the current literature.

R. We acknowledge this comment. The subject of agriculture 4.0 is very extensive and at the moment acquired great interest, in the present work these technologies are mentioned since the neural networks can serve as tools to these. These topics were added because we want to present the panorama of the use of neural networks in agriculture 4.0 and greenhouses. However, the working group does plan to develop a full article focusing on agriculture 4.0 and 5.0, covering the aforementioned topics.

Round 2

Reviewer 2 Report

Authors have restructured the paper and added relevant diagrams for modelling and describing types of artificial neural networks. Author has added satisfactory changes to the second version of manuscript along with the addition of suggested content. Please consider the following remarks-

  • Change the caption for Figure 1, you have mentioned “Greenhouses and approach to describe them”, whereas, in the diagram you are just classifying based on models.
  • Cross check, all the diagrams are self-explanatory.
  • Proof reading is required to bring the consistency w.r.t grammar and punctuations throughout the paper.
  • In the abstract, author claims to provide the guidelines for developers which is missing throughout the paper. Also, it is not mentioned in the conclusion. It would be nice to add a Section i.e Guidelines/Discussion after Section 6 to suggest possible approaches and research roadmap w.r.t applications.
  • In the introduction, author claim to show the trend for future research in the development of the model. But this is not covered well in the paper. Include this as well in the aforementioned remark.
  • Promising technologies such as IoT, ML, big data are covered too superficially without SOTA and its possible integration with the discussed models.

Author Response

Reviewer 2

Authors have restructured the paper and added relevant diagrams for modelling and describing types of artificial neural networks. Author has added satisfactory changes to the second version of manuscript along with the addition of suggested content. Please consider the following remarks:

  • Change the caption for Figure 1, you have mentioned “Greenhouses and approach to describe them”, whereas, in the diagram you are just classifying based on models.

R. We acknowledge this comment. We adjust the title and the figure as shown:

Figure 1. Interest topics in greenhouses and models classification

  • Proof reading is required to bring the consistency w.r.t grammar and punctuations throughout the paper.

R. We acknowledge this comment. A revision of the document was carried out and punctuations and grammar were corrected

  • In the abstract, author claims to provide the guidelines for developers which is missing throughout the paper. Also, it is not mentioned in the conclusion. It would be nice to add a Section i.e Guidelines/Discussion after Section 6 to suggest possible approaches and research roadmap w.r.t applications. In the introduction, author claim to show the trend for future research in the development of the model. But this is not covered well in the paper. Include this as well in the aforementioned remark.

R. We acknowledge this comment. Added a new section entitled "Guidelines for the application of neural networks in greenhouses" in which the elements indicated are addressed as shown below (lines 1204-1239):

In this review we have presented the application of ANNs in the prediction of the microclimate in greenhouses, their use in energy optimization, as well as other applications in greenhouses. Of these topics addressed, it should be noted that the potential of the ANNs continues to be promising for future research. Although studies have presented ways in which one should delve further, such as the use of physical models in conjunction with neural networks, the work is still few.

Physical models for the creation of synthetic data is a good strategy to feed neural networks, since it would complement in situ data and would allow confronting the possible problems that the nature of this type of data entails. Linker and Seginer [208] presents two possible configurations, in series and in parallel. However, serial configuration has not been tested and its use in conjunction with statistical tools such as PCA would optimize the data selection process for a neural network. Also, the structure of hybrid networks could be expanded and the serial and / or parallel configurations could be used in conjunction with optimization algorithms such as GA, PSO, among others.

The use of WSN in greenhouses is part of the SA, offering advantages not only in data collation and greenhouse climate control, but also in the energy consumption of this type of device. The application of the ANNs in the WSN of the greenhouses would also be an interesting topic to develop, since on the one hand there is the automatic reprogramming of the wireless sensors and on the other the forecast of the greenhouse variables collated using this sensor network.

RNNs are little used compared to FFNNs, their use in greenhouses in conjunction with image analysis can be of help in the identification of diseases and pests in greenhouses. Also, DL and CNNs are tools that would facilitate these tasks. Likewise, the forecast of the microclimate in greenhouses using analysis of thermographic images and CNNs would be something interesting to apply.

The CNNs in the study of growth and transpiration crops is an issue that would be worth developing. In traditional methods, these processes represent a mathematical challenge while CNNs would simplify them with help of morphological and thermal analysis.

The application of new technologies such as 4.0 in greenhouses opens the panorama of carrying out work with a perspective on integration and exchange information. The studies presented have been developed with data obtained from a single greenhouse that is, the object of study has been the case of a particular greenhouse. However, with the IoT, the WSN, the CC, among others, information from various points (greenhouses) could be accessed at any time. ANNs can be developed that use this data, but first it would be necessary to make a reliable database. Although each greenhouse is in the same region or has a similar outside climate, variations would still be present. However, synthetic data could be used to minimize these variations. The WSN could be trained to detect diseases and plagues in crops and this information can be used in nearby greenhouses to predict the presence of these affectations and take the necessary preventive measures.

In addition, the following conclusion was added (lines 1260-1262):

An important guideline for future works is the integration and exchange of information using 4.0 technologies. The role of the ANNs is to develop predictive models that take advantage of the information generated and its management.

  • Promising technologies such as IoT, ML, big data are covered too superficially without SOTA and its possible integration with the discussed models.

R. We acknowledge this comment. The subject of agriculture 4.0 is very extensive and at the moment acquired great interest, in the present work these technologies are mentioned since the neural networks can serve as tools to these. These topics were added because we want to present the panorama of the use of neural networks in agriculture 4.0 and greenhouses. However, the working group does plan to develop a full article focusing on agriculture 4.0 and 5.0, covering the aforementioned topics.

Reviewer 3 Report

The Authors significantly improved the manuscript corresponding to reviewer's comments, and presented satisfactory explanations. Now the paper can be published in Applied Sciences.

Actually I am not satisfied that the Authors have kept a description of a structure and topology of a general ANN, as such information is given in many textbooks. However, the Authors explained this situation - it was an intention of one of the reviewers to present that part on material.

I would like to emphasize that the Authors analyzed implementation of a variety of topologies of artificial neural networks for greenhouses, a variety of greenhouse microclimat parameters and, above all, they presented a comprehensive list of input variables confronted with output variables. This is definetely a valuable part and a strong point of the paper.

Author Response

Thank you very much for the comments, some changes were made to the article. Changes are highlighted in yellow and one more topic has been added. Grammar and punctuation were revised.

This manuscript is a resubmission of an earlier submission. The following is a list of the peer review reports and author responses from that submission.

Round 1

Reviewer 1 Report

This paper summarizes the use of artificial neural networks for greenhouse microclimate prediction and control and energy consumption optimization. While the topic is of great interest for its applications in agriculture, the paper needs substantial revision. The paper summarizes main concepts of NN and its applications but doesn’t critically review the literature to address in detail why this model is a good model for greenhouse microclimate, what are the main advantages and disadvantages, challenges or future directions. Therefore, I recommend rejection.

Suggestions (not exhaustive) :

- As the most important comment, the review doesn’t address why NN are a good model to predict the greenhouse microclimate that is "complex, multiparametric, nonlinear and depends on a set of external and internal factors". What properties of NN handle such characteristics? Is the data available enough to train a NN for this? How much data would a greenhouse need? How does NN compare to other methods both in terms of accuracy and costs?  What will be the main challenges for this area based on the review?

- The review summarizes NN concepts but fails to summarize microclimate concepts. Moreover, the whole discussion about NN feels disconnected from the main topic, a general summary not adapted to the audience of the paper. How would you model microclimate with a NN? The application section doesn’t address this points in sufficient detail.  Moreover, the discussion about physical neurons could be left out. It is better to explain why the NN can model microclimate conditions. In addition, there is no explanation of the microclimate of greenhouses. Given the broad audience of the journal, I would recommend to include a brief description, reducing the description of NN.

- Many arguments do not have enough support. For example, in line 2643“The objective of using an RBFANNs is that the design and training process is a simpler task. “ in line 408 “however, the use of other types of models such as those based on neural networks is a more efficient option “ In line 656 The results obtained with the ANNs compared to physical models are better,” -> how much better? Is it significant?

Some Grammar Suggestions:

- Don’t use reference numbers as the subject for a sentence.

- Be careful with runaway phrases. For example: line 71: “In comparison to models obtained by analyzing the input and 71 output of process data, or black-box models. “

Author Response

Reviewer 1

Suggestions (not exhaustive):

Q1. Why NN are a good model to predict the greenhouse microclimate that is "complex, multiparametric, nonlinear and depends on a set of external and internal factors".

R. We acknowledge this comment. We added this information in L 385-388 as follows:

Neural networks application in the microclimates control is a topic that has currently gained interest, NN provides reliable models that can reflect the non-linear characteristics of the greenhouse that are difficult to solve using traditional techniques, and does not require any prior knowledge of the system and is very suitable for modeling dynamic systems in real time [77,78].

Q2. What properties of NN handle such characteristics?

R. Comment acknowledged. We mentioned this information in the L 109-118. as follows:

An artificial neural network is a machine learning algorithm based on the concept of a human neuron [50]. It is a biologically inspired computational model, consisting of processing elements (neurons) and connections between them with coefficients (weights) attached to the connections [51]. As mentioned, ANNs are inspired by the brain structure for this reason it is important to define the main components under which a neuron, dendrites, cell body, and axon works (Figure 1).

Q3. Is the data available enough to train a NN for this?

R. We acknowledge this comment. We added this information in L 118-120 The team is currently developing a neural network using 70032 data collected during 1 year, of which 50% will be used for network training, 25% for verification and 25% for testing.

Q4. How much data would a greenhouse need?

R. Some studies propose that to have an adequate representativeness of the source data it must contain at least one year of measurements [133]

Q5. How does NN compare to other methods both in terms of accuracy and costs? 

R. We acknowledge this comment. We added this information in L 75-82 as follows:

Prediction methods can be divided into two groups, physical methods based on mathematical theory, which require a large number of parameters to be determined, as well as the difficulty of measuring those parameters; black box methods based on modern computational technology (particle swarm optimization algorithm, least squares support vector machine model), which do not always guarantee convergence to an optimal solution and easily undergo partial optimization [47]. On the other hand, instead of being programmed, neural networks learn to recognize patterns. These systems are highly appropriate to reflect knowledge that cannot be programmed or justified, as well as to represent non-linear phenomena [48].

Q6. What will be the main challenges for this area based on the review?

R. The main challenges for this area presented in the topic Perspectives: Greenhouse artificial neural networks application L 743-791

Q7. Moreover, the whole discussion about NN feels disconnected from the main topic, a general summary not adapted to the audience of the paper. How would you model microclimate with a NN? The application section doesn’t address these points in sufficient detail.  Moreover, the discussion about physical neurons could be left out. It is better to explain why the NN can model microclimate conditions. In addition, there is no explanation of the microclimate of greenhouses. Given the broad audience of the journal, I would recommend to include a brief description, reducing the description of NN.

R. We acknowledge this comment. We added a section (L 385-421) as follows:

3.1.      Greenhouse microclimate

The application of neural networks in the control of microclimates is a topic that has currently gained interest, NN provides reliable models that can reflect the non-linear characteristics of the greenhouse that are difficult to solve using traditional techniques, and does not require any prior knowledge of the system and is very suitable for modeling dynamic systems in real time [77, 78]. The greenhouse microclimate refers to the environmental conditions that plants require to be in good condition. Cultivation represents central and the most complex element of the system. Due to this complexity and the great diversity of greenhouse crops, it is common to consider only certain general questions that are most relevant to the response of crops in relation to the microclimate. The relevant results are total quantity produced, the start and production time, and the quality of the product, all of which influence crop yield. The main factors that characterize the greenhouse climate are the concentration of CO2, air temperature and humidity. In most cases, radiation is imposed by external weather conditions and is generally considered to be an external condition. Likewise, due to the transpiration, photosynthesis and respiration processes, the crop interferes with the CO2 and water vapor mass and energy balances in the greenhouse air. For this reason, the harvest plays a double role: it modifies and responds to its environment [79]. Greenhouse interior generally behaves like a multivariable nonlinear system. It is for this reason that the microclimate is considered as a homogeneous block, which means that the indoor air is well mixed. A greenhouse mainly includes the following five classes of variables: 1) Variables of the state of the greenhouse microclimate: Air temperature, air humidity and CO2 concentration. 2) Relevant variables for the crop growth: Photosynthesis rate, transpiration rate, respiration rate, leaf area index, crop dry matter and fruit dry matter. 3) External climatic variables: Outdoor air temperature, outdoor air humidity, solar radiation, speed and wind direction. 4) Control input variables: Heating, ventilation, CO2 injection, misting, among others. 5) Physical parameters of the material and structure: Floor area, height, plant density, among others [80].

Temperature and humidity are of the most relevant parameters in the greenhouse microclimate, since they have complex exchanges and interactions of heat and mass between the inner air, other elements of the greenhouse and the outside. Building a model is a difficult assignment with simple mathematical formulas or transformation functions. However, the method of building models with ANNs has a great capacity for mapping nonlinear functions, which is applied to many production process systems [81]. For the network design, air temperature and humidity of the greenhouse air are generally considered as outputs, this due to the aforementioned. However, setting the inputs is more complicated and requires an understanding of the system. It is not convenient to consider a large number of inputs, as this could cause uncontrolled extrapolations instead of increasing the estimation power. Three elements can be considered in order to consider an input variable: 1) correlation of selected input with other inputs, 2) physical dependence nature of the output and the input 3) input variable range. Third point, solar radiation at greenhouse temperature can be considered as an example [82].

Q8. Many arguments do not have enough support. For example, in line 2643“The objective of using an RBFANNs is that the design and training process is a simpler task. “in line 408 “however, the use of other types of models such as those based on neural networks is a more efficient option “In line 656 The results obtained with the ANNs compared to physical models are better,” -> how much better? Is it significant?

R. We acknowledge this comment. We support this information.

Q9. Some Grammar Suggestions:

R. We acknowledge this comment. We checked grammar manuscript.

Q10. Don’t use reference numbers as the subject for a sentence.

R.We acknowledge this comment. We change the references by authors.

Q11. Be careful with runaway phrases. For example: line 71: “In comparison to models obtained by analyzing the input and 71 output of process data, or black-box models. “

R. We acknowledge this comment. We added change information in L 75-82.

Reviewer 2 Report

The manuscript (MS) reviews the applications of artificial neural networks (ANNs) in predicting microclimate variables and energy consumptions of greenhouse systems. The authors also provide their perspectives on the future integration of ANNs with other technologies for smart agriculture development. While the work presented a thorough literature review of ANN applications related to greenhouse systems design, the manuscript requires a revision before it can be published in Applied Science for the following reasons:

  1. It is not clear what the main targeted audience of the MS is.

On the one hand, if the MS targets a general audience with a little background on ANNs and machine learning algorithms, it should attempt to better define ANNs and related concepts. I found section 2 quite long and confusing. The authors should give a concise definition of ANNs with good citations. For example, “ANNs (Hopfield, 1982; Kohonen, 1982) is a machine learning technique that mimics biological nervous systems, such as the brain, to identify patterns (classification) or to fit functions (regression)”. The authors should also make sure that they clearly explain different basic concepts such as the five basic elements of an elementary artificial neuron. If the authors want to compare the similarities and differences between how an ANN and a brain work, they can consider including a visual presentation of a nervous system to better depict different nervous components (e.g., neuron, dendrites, cell body, and axon). Likewise, the authors can better explain activation functions, type of activation functions and when to use them.

On the other hand, if the authors aim at communities with some knowledge of machine learning techniques, they should attempt to highlight the differences between feed-forward and recurrent ANNs when reviewing ANNs applications. For instance, it is important to review/comment on the advantages and disadvantages (e.g., in terms of accuracy, training/prediction time, complexity) of different ANNs, training methods, activation functions used in different aspects of greenhouse design studies. Since the authors support the use of complex ANNs algorithms such as deep learning and hybrid ANNs in future applications, they should better argue the necessity and the trade-offs of such applications. When discussing Agriculture 4.0, some concepts such as IoT, PA, AI, CC should be clearly explained for readers since they are still quite new.

  1. The grammar and writing of the MS need to be improved.

I found the writing wordy with many long and complex sentences. There are many clumsy expressions throughout the MS. For example, “To minimize this cost function, several methods are used, is the gradient descent as the error back-propagation (BP) algorithm the most used for its acceptable results in one layer and multilayer networks” (lines 242-244) or “The application of methods and tools that simplify the treatment of the variables related to the climate of the greenhouses is a very important issue, since the speed of calculation, the precision in the prediction of the behavior and control of the variables of the different elements follow presenting a significant challenge” (lines 249-252). I would recommend the authors spend more time on rephrasing these sentences. It is also good to have the MS proof-read by a native English editor. Some of the abbreviations such as DL, DNN, LSTMNNs were not defined prior to their usage in the text. The MS is also full of complex abbreviations, so I would recommend using full texts for some ANN-unrelated concepts such as smart agriculture and precision agriculture to increase readability.

Tables 2, 3, and 4 should use smaller font sizes with single spacing. The author should consider adding other studies that they discussed in the MS text into the table. For example, table 2 should also include results from [38], [47], [63] – [66], and table 3 should have results from [74] – [77]

Author Response

Reviewer 2

Q1. On the one hand, if the MS targets a general audience with a little background on ANNs and machine learning algorithms, it should attempt to better define ANNs and related concepts. I found section 2 quite long and confusing. The authors should give a concise definition of ANNs with good citations. For example, “ANNs (Hopfield, 1982; Kohonen, 1982) is a machine learning technique that mimics biological nervous systems, such as the brain, to identify patterns (classification) or to fit functions (regression)”.

R. We acknowledge this comment. We added a clearly definition as follows: (L 109-118)

An artificial neural network is a machine learning algorithm based on the concept of a human neuron [50]. It is a biologically inspired computational model, consisting of processing elements (neurons) and connections between them with coefficients (weights) attached to the connections [51]. As mentioned, ANNs are inspired by the brain structure for this reason it is important to define the main components under which a neuron, dendrites, cell body, and axon works (Figure 1).

Q2. The authors should also make sure that they clearly explain different basic concepts such as the five basic elements of an elementary artificial neuron.  If the authors want to compare the similarities and differences between how an ANN and a brain work, they can consider including a visual presentation of a nervous system to better depict different nervous components (e.g., neuron, dendrites, cell body, and axon).

R. We acknowledge this comment. We include a Figure 1. Structure of a biological neuron (explication is in L 120-128)

Q3. Likewise, the authors can better explain activation functions, type of activation functions and when to use them.

R. We acknowledge this comment. We re-wrote activation functions explain as follows (L181-182):

The activation function is a function that receives an input signal and produces an output signal after the input exceeds a certain threshold. That is, neurons receive signals and generate other signals [56].

We wrote type of activation functions and when to use them explanation as follow (L187-197):

The sigmoid function is the most used activation function in ANNs. An important feature of sigmoid function is that it is differentiable, whereas binary function is not. Also, this function varies from 0 to +1. Sometimes it is sought that the activation function oscillates between –1 and +1, in which case the activation function assumes an antisymmetric form with respect to the origin, defining it as the hyperbolic tangent function [58, 59]. The hyperbolic tangent function has a similar response to the sigmoid function inputs, but they differ in the output ranges [60].

The linear trigger function will only produce positive numbers in the entire range of real numbers. Whereas, the piecewise linear activation function is also called the linear saturation function and can have a binary or bipolar range for the saturation limits of the output. And the Gaussian activation function can be used when finer control over the activation range is needed [61].

Q4. On the other hand, if the authors aim at communities with some knowledge of machine learning techniques, they should attempt to highlight the differences between feed-forward and recurrent ANNs when reviewing ANNs applications.

R. We acknowledge this comment. We completed this information L 341-356 as follows:

RNN is distinguished from a FFNN by the presence of at least one feedback connection. FFNNs do not have the intrinsic ability to process temporary information. There are two important considerations about why recurrent networks are viable tools for modeling: inference and prediction in noisy environments. In a typical recurrent network architecture, the activation functions of the hidden unit are fed back at each time step to provide additional input. That is, the recurrent networks are built in such a way that the outputs of some neurons feed back to the same neurons or to the neurons in the previous layers [70]. Feedback from hidden units allows filtered data from the previous period to be used as additional input in the current period. This causes the network to work not only with the new data, but also with the past history of all entries, as well as their leaked equivalents. This additional filtered input history information acts as an additional guide to assess the current noisy input and its signal component. By contrast, filtered history never enters a FFNN. This is where recurring networks differ from a FFNN. Second, since recurrent networks have the ability to maintain the past history of filtered entries as additional information in memory, a recurrent network has the ability to filter noise even when the noise distribution can vary over time. Whereas in a FFNN a completely new training must be carried out with a new data set containing the new type of noise structure [73].

Q5. For instance, it is important to review/comment on the advantages and disadvantages (e.g., in terms of accuracy, training/prediction time, complexity) of different ANNs, training methods, activation functions used in different aspects of greenhouse design studies.

R. We acknowledge this comment. We completed this information in 2.2.1 and 2.2.2 sections.

Q6. Since, the authors support the use of complex ANNs algorithms such as deep learning and hybrid ANNs in future applications, they should better argue the necessity and the trade-offs of such applications.

R. The main challenges for this area presented in the topic Perspectives: Greenhouse artificial neural networks application L 743- 791.

Q7. When discussing Agriculture 4.0, some concepts such as IoT, PA, AI, CC should be clearly explained for readers since they are still quite new.

R. We acknowledge this comment. We re-wrote the section 6 as follows: (Changes are highlighted)

6.1 Artificial Intelligence

Artificial Intelligence (AI) researches and builds intelligent software and machines, provides a particular solution to a particular defined complex problem, is made up of branches such as genetic algorithms, particle swarm optimization, simulation and artificial neural networks and hybrid models (two or more of the above) [120, 121]. AI consists of mapping non-linear behavior between inputs and outputs of processes [122]. AI consists of a large number of practical tools that allow solving difficult problems tasks that require biological or human intelligence, with functions such as perceptron, recognition, Decision-making and control combines brain science and related fields, such as cognitive science and psychology [123]. The AI allows the prediction of thermal properties of biomass, tools such as ANN have proven to be vital in the development of research in the prediction of biomass energy, which in turn could be used in the control of greenhouse microclimates. NNs are flexible to accommodate non-linear and non-physical data, however they require large multidimensional data sets to reduce the risk of extrapolation. [124]. AI employs quite different mathematical and algorithmic approaches, from operational research to restricted programming, DL and ML [125].

DL expands on classic ML by adding more depth to modeling. Its advantage is feature learning, that is, automatic extraction of features from raw data and quick resolution of complex problems. The DL is made up of various components, such as convolutions, grouping layers, fully connected layers, gates, memory cells, activation functions, encoding / decoding schemes, depending on the network architecture used, such as the mentioned convolutional neural networks (CNN) [126].

6.2 Future of deep learning in greenhouse agriculture

Deep learning has demonstrated a great capacity in pattern recognition and machine learning. Among One of the main tasks of this type of network is to learn to actively perceive patterns by sequentially directing attention to relevant parts of the available data [119]. The advantage of DL over conventional networks is the possibility of developing simulated data sets to train the model, which would allow solving real-world problems, such as greenhouse systems. In [127] the various applications of DL in agriculture are exposed, however, its use in greenhouses is still lacking. Of the studies that have been carried out of DL in greenhouses we can find the one carried out in [126], who propose a new deep RNN, with a Long Short-Term Memory neural network (LSTMNN)  model to predict the stem diameter, or tomato performance problems using environmental parameters such as CO2, humidity, radiation, outside temperature and indoor temperature. One of the main disadvantages of this method is exposed, the large amount of data necessary for the training process.

The BP network is the basis of the vast majority of DL algorithms, it also allows models that consist of multiple layers of processing to represent data with multiple levels of abstraction, so its application in agriculture has begun to be studied. In the field of smart farming (SF) it can be used for the detection of plant diseases, weed control, and plant counting through image recognition. Convolutional Neural Networks (CNN), RNNs, and Generative Adversarial Networks (GANs) being the most viable types of deep networks in this field [128].

The advantage of DL over conventional networks is the possibility of developing simulated data sets to train the model, which would allow solving real-world problems, such as greenhouse systems. In [100] the various applications of DL in agriculture are exposed, however, its use in greenhouses is still lacking. Of the studies that have been carried out of DL in greenhouses we can find the one carried out in [101], who propose a new deep RNN, with a Long Short-Term Memory (LSTM) neural model to predict the stem diameter, or tomato performance problems using environmental parameters such as CO2, humidity, radiation, outside temperature and indoor temperature. Where one of the main disadvantages of this method is exposed, the large amount of data necessary for the training process.

CNN models are an extension of the DL. They consist of MLP networks that involve multiple pools and fully connected layers, learn and optimize filters on each layer through the back-propagation mechanism. These trained and learned filters extract features that distinctively represent the input image. This type of model has managed to overcome state-of-the-art algorithms, since then it became the most advanced method in many data processing tasks. Currently, CNN architectures are trained from scratch or adjusting pre-trained architectures. Using pre-trained architectures allows transfer learning to be used. Transfer learning consists of using the learning of models that have been previously trained with large data sets from other systems, in other problems or similar systems [129].

CNN has a great capacity in image processing, which makes it widely used in agricultural research. The challenge with the use of information is to interpret the collected images. Interpreting satellite images using CNN and GA has become a useful decision-making strategy, especially for precision agriculture (PA) [130]. Furthermore, they can also be used in weather forecasting, which is key for agriculture [131].

6.3 Future of hybrid ANNs in greenhouse agriculture

The use of the combination of ANNs with mathematical models has been little explored, however, as can be seen in Yousefi et al. [132] and Linker et al. [133], the approach path can be considered from two perspectives: First, using relying on techniques such as as the fuzzy logic for optimization in the random choice of the initial parameters and second, to or use the physical models for the generation of synthetic data that help the network in the learning process, minimizing the errors due to the lack of information that a base can present of data in situ. ANN hybrid models have the potential to provide forecasts that work well compared to more traditional modeling, such as the use of ANN models optimized by particle swarm (PSO) and genetic algorithms (GA) that have shown good prognostic results of energy requirements [134].

6.4 Agriculture 4.0 and the ANNs

Nowadays, farmers need to adapt to new technologies and apply them in agriculture. The concept of the internet has evolved in recent years, from a focus on hardware to one on services and applications, coupled with this, the development of precision agriculture (PA) is increasingly important due because of the increase in demand for food and the impacts of climate change. Agriculture 4.0 is the integration of technologies (IoT, PA, AI, CC among others) for automating tasks and cyber-physical systems, allowing production planning and control [135].

6.4.1 Precision agriculture and Internet of Things

The PA integrates the new technologies derived from the information age with the agricultural industry. It consists of a crop management system that tries to optimize the type and quantity of inputs with the real needs of crops for small areas within an agricultural field. PA uses crop inputs more effectively, including fertilizers, pesticides, tillage, and irrigation water [136].

As a management tool, PA consists of five elements: geographic positioning (GPS), information gathering, decision support, variable-rate treatment, and performance mapping. Yield mapping allows the farmer to monitor the actual result of the different inputs, being a tool for collecting information on previous years. For this reason, large data sets (Big Data) are required to interpret specific variables. In this area, new technologies are still under development [137]. Mapping many different factors of soil, crops, and the environment produces large amounts of data. Farmer data overload must be overcome by integrating expert systems and decision support systems [138], which in turn must be based on models such as those that have been exposed throughout this writing.

AP has been applied and developed in greenhouses [139-141], as well as the use of neural networks as a support tool [142]. Being the real-time monitoring systems for the management of the greenhouse to control environmental parameters, the area under which it is necessary to go deeper [143]. Likewise, smart agriculture (SA) broadens the concept of PA, since the tasks for decision management are reinforced by knowledge of the situation. This in turn causes real-time assistance resources to be required to perform agile actions such as the Internet of Things (IoT) [144].

IoT is the interaction between a variety of physical things or objects that use specific addressing schemes to connect to the Internet, this type of technology allows the inherent reduction of environmental impact by real-time reaction to alert events such as detections of weeds, pests or diseases, climate or soil monitoring warnings, which allow a reduction and the adequate use of inputs such as agrochemicals or water [145, 146].

One of the advantages of IoT is its ability to control other devices remotely transversely based on the existing system, which makes a good interrelation between the physical world and different computer-based frameworks and creates possibilities for greater financial effectiveness advantage and precision. In the near future, IoT is trusted to give numerous more administrations such as [147]]. Currently, IoT have been implemented in crop care. Kitpo et al [148] applied IoT to determine the date of tomato harvest, for this they carried out a monitoring of the 6 different stages of tomato cultivation, using as parameter the visible wavelength as a characteristic in the classification of support vector machines (SVM). Climatic data such as temperature, humidity, illuminance, among others, were recorded daily during tomato cultivation, these data and the data obtained from the SVM classification were used for the training of a NN, the results applied to the elaboration of an automated system by using IoT to support greenhouse growers in the future.

Tervonen [149] studied the effectiveness of IoT in quality control during vegetable storage. During the storage of potatoes, it determined that for the proper control of temperature and other parameters, multiple measurement points are required in different locations to guarantee the desired behaviors for the entire volume. Wang et al. [150] verified that the data loss rate between the data acquisition unit and the gateway was 1.52%, and the data loss rate was 0.4% between the gateway and the server, making the IoT system feasible for monitoring greenhouses. IoT has emerged as an alternative for optimizing the agricultural sector since it allows farmers to monitor their agricultural fields in real time and receive recommendations to produce good quality crops while maximizing their overall profits on the products sold [151]. Linked to the IoT, there is the Cloud computing (CC). CC is a model that allows convenient access to the network request to share configurable calculation resource groups [152], it is a model to allow ubiquitous, convenient network access and on demand to a shared pool of computing resources that can be quickly provisioned and released with minimal effort from management or service provider interaction [153].

Q8. Tables 2, 3, and 4 should use smaller font sizes with single spacing. The author should consider adding other studies that they discussed in the MS text into the table. For example, table 2 should also include results from [38], [47], [63] – [66], and table 3 should have results from [74] – [77]

R. We acknowledge this comment. We completed Table 2 and 3. We change font size and single spacing of table 2, 3 and 4.

Reviewer 3 Report

Dear Authors,

thank you very much for your work. The theme is relevant and you did a good review. However, there are several not clear points which need corrections.

The statement in line 71 is not clear:"In comparison to models obtained by analyzing the input and 71 output of process data, or black-box models"

It is not clear what is total yield in line 78? Not clear what does it means.

There is something I did not understand in line 97: you wrote "incomplete information", but it sonds strange. ANN and ML in a general aspect is relevant when such problem has unknown rules or equations. Incomplete information is a term usually identifies an error. Could you clarify what is the meaning of your statement?

Line 112 and 113 could be changed (single neuron, each neuron...)

It would be interesting to discuss a bit better what is a activation function (line 138, 139 and table 1). You present a threshold limit, which is valid for the binary step function. However, taking into consideration the other functions how can you support the threshold argument? I believe you should rethink the argument.

Item 2.2.1 needs a review since it is written the information flows in just one direction, however, in line 263 and 264 it is written that information flows back (MLP). I suppose that I  understood your argument, but it must be clear (how to argument back propagation with data flow in just one direction?)

Item 2.2.2 line 197 - an example of RNN is a network that is not a RNN?

In item 3.1 what does it mean online and offline training? what's the difference in the training process?

In table 2 (reference 70) it seems to be a misunderstanding: there is one parameter that is the input and output. Is it correct? In this case what the effect in the ANN?

In item 3.2 - table 3 is not clear and I did not find nothing that supports RNN is lower than FFNN. What is lower? training or testing? If it was related to training, what is the importance? What is the impact?

In line 319 what is good performance? It is not clear and with this information it is not possible to evaluate results without indicating numbers (in line 465 you did this). Same need to be corrected in line 369 and 370.

Table 4 has the same situation as table 2: same input and output - it seems something inconsistent. In this situation the result of the ANN is considered trivial and the comments in the table does not make sense.

In line 414 it is not clear the 95%. Is it better then regression model or 95% is the NN accuracy?

What is load in line 415?

In line 428 what is 1sec? Training or testing?

In the item 6 it would be interesting a comparation between all algorithms and results presented in the article. It would improve the understanding the topic. It is important to evaluate all the presented results.

I hope such comments could help you the article improvements.

Author Response

Reviewer 3

Q1. The statement in line 71 is not clear: "In comparison to models obtained by analyzing the input and output of process data, or black-box models"

R. We acknowledge this comment. Lines 75-82 were re-wrote as follows:

Prediction methods can be divided into two groups: physical methods based on mathematical theory, which require a large number of parameters to be determined, as well as the difficulty of measuring those parameters; and black box methods based on modern computational technology (particle swarm optimization algorithm, least squares support vector machine model), which do not always guarantee convergence to an optimal solution and easily undergo partial optimization [47]. On the other hand, instead of being programmed, neural networks learn to recognize patterns. These systems are highly appropriate to reflect knowledge that cannot be programmed or justified, as well as to represent non-linear phenomena [48].

Q2. It is not clear what is total yield in line 78? Not clear what does it means.

R. We acknowledge this comment. Line 89-90 was re-wrote as follows:

 …. and simplify the structure of the network and in this way improve greenhouse total yield [49].

Q3. There is something I did not understand in line 97: you wrote "incomplete information", but it sounds strange. ANN and ML in a general aspect is relevant when such problem has unknown rules or equations. Incomplete information is a term usually identifies an error. Could you clarify what is the meaning of your statement?

R. We acknowledge this comment. Lines 109-112 were re-wrote as follows:

An artificial neural network is a machine learning algorithm based on the concept of a human neuron [50]. It is a biologically inspired computational model, consisting of processing elements (neurons) and connections between them with coefficients (weights) attached to the connections [51].

Q4. Line 112 and 113 could be changed (single neuron, each neuron...)

R. We acknowledge this comment. Line 152 was re-wrote as follows:

Apiece neuron ….

Q5. It would be interesting to discuss a bit better what is an activation function (line 138, 139 and table 1). You present a threshold limit, which is valid for the binary step function. However, taking into consideration the other functions how can you support the threshold argument? I believe you should rethink the argument.

R. We acknowledge this comment. Line 181-197 was re-wrote as follows:

The activation function is a function that receives an input signal and produces an output signal after the input exceeds a certain threshold. That is, neurons receive signals and generate other signals [56]. The neuron start is only performed when the sum of the total inputs is greater than the neuron threshold limit, then the output will be transmitted to another neuron or environment [57]. This threshold limit determines whether or not the neuron is activated or not, the most common activation or transfer functions are the linear functions, binary step, piecewise linear, sigmoid, Gaussian and hyperbolic tangent functions (Table 1). The sigmoid function is the most used activation function in ANNs. An important feature of sigmoid function is that it is differentiable, while binary function is not. Also, this function varies from 0 to +1. Sometimes it is sought that the activation function oscillates between –1 and +1, in which case the activation function assumes an antisymmetric form with respect to the origin, defining it as the hyperbolic tangent function [58,59]. The hyperbolic tangent function has a similar response to the sigmoid function inputs, but they differ in the output ranges [60].

The linear trigger function will only produce positive numbers in the entire range of real numbers. Whereas, the piecewise linear activation function is also called the linear saturation function and can have a binary or bipolar range for the saturation limits of the output and the Gaussian activation function can be used when finer control over the activation range is needed [61].

Q6. Item 2.2.1 needs a review since it is written the information flows in just one direction, however, in line 263 and 264 it is written that information flows back (MLP). I suppose that I understood your argument, but it must be clear (how to argument back propagation with data flow in just one direction?)

R. We acknowledge this comment. This section was re-wrote as follows: (L 226-255)

As mentioned above, neuron is the basic component of neural networks. Neurons are connected to each other through synaptic weight. Considering a neural network with three layers: an input layer, a hidden layer and an output layer the intermediate layer is considered as a self-organized Kohonen map, which consists of two layers of processing units (input and output), depending on the complexity of the network, there may be several hidden layers in each network (Figure 3). In Feedforward neural networks, information progresses, from the input nodes to the hidden nodes and from the hidden nodes to the output nodes. When an input pattern is fed into the network, the units in the output layer compete with each other, and the winning output unit is the one whose input connection weights are closest to the input pattern, the number of Neurons in the input and output layers is the same as the number of inputs and outputs of the problem. The proposed learning method consists of two stages, the first stage is to determine the neuron of the hidden layer whose weight vector is the first input vector and the second refers to the training process. Initially, the Euclidean distance between the input and the weight vector of the first neuron will be calculated. If the distance is greater than a predetermined distance threshold value, a new hidden-layer neuron is created by assigning the input as the weight vector. Otherwise, the input pattern belongs to this neuron. During training, each pattern presented to the network selects the closest neuron on a Euclidean distance measure, modifying the winner's weight vector, and topological neighbors draws them in the direction of the input, the weights leaving the winning neuron and its neighbors are adjusted by the gradient descent method. Forward neural networks fall into two categories based on the number of layers, either single layer or multiple layers. [63-66][55].

Backpropagation is a type of ANN training, used to implement supervised learning, tasks for which a representative number of sample inputs and correct outputs are known. Backward propagation is derived from the difference in desired and predicted, output; this is calculated and propagated backward. First network weights to a small random weight are initialized, the vector set of input data to the network are presented, propagate the input to generate the output, which is called the input advance phase, calculate the error comparing the estimated net output with the desired output. The weights will be corrected from the output to the input layer, that is, in the backward direction in which the signals propagate when objects are introduced into the network. This is repeated until the error no longer improves [67-69].

Q7. Item 2.2.2 line 197 - an example of RNN is a network that is not a RNN?

R. We acknowledge this comment. We change this sentence (L 295-297) as follows:

Hopfield network can function as a solid memory and resistant to the alteration of the connection (Figure 4b). There is a guarantee in terms of convergence for this network; however, it is not a recurring network in general [71].

Q8. In item 3.1 what does it mean online and offline training? what's the difference in the training process?

R. We acknowledge this comment. We added this information as follows (L 441-445):

Offline learning consists in adjust to weight vectors and network thresholds after the entire training set is presented (requires at least one training data stage), while in online learning network weights and thresholds adjustments are made after each training sample is submitted (after executing the adjustment step, sample can be discarded)[85].

Q9. In table 2 (reference 70) it seems to be a misunderstanding: there is one parameter that is the input and output. Is it correct? In this case what the effect in the ANN?

R. We acknowledge this comment. All the inputs correspond at outside parameters while all output correspond only inside parameters.

Q10. In item 3.2 - table 3 is not clear and I did not find nothing that supports RNN is lower than FFNN. What is lower? training or testing? If it was related to training, what is the importance? What is the impact?

R. We acknowledge this comment. An error is presented in the wording of the description in Table 3. The objective of said table is to show the different studies that have been carried out on the ANN, however it is not a comparative table with the RNN.

We re-wrote line 502 as follows: Compared to FFNN effort, the application of RNN is a less studied field

Q11. In line 319 what is good performance? It is not clear and with this information it is not possible to evaluate results without indicating numbers (in line 465 you did this). Same need to be corrected in line 369 and 370.

R. We acknowledge this comment. We added numeric values as follows (L 508-509) the operation and control of the greenhouse, they connected both neural networks in cascaded obtaining a lower criterion error (Ec= 344.12) in comparison of a neural network simple (Ec= 533.31)

L 584-585. The models showed good performance (daily average absolute simulation error smaller than 1â—¦C) for long periods without

Q12. Table 4 has the same situation as table 2: same input and output - it seems something inconsistent. In this situation the result of the ANN is considered trivial and the comments in the table does not make sense.

R. We acknowledge this comment. All the inputs correspond at outside parameters while all output correspond only inside parameters.

Q 13. In line 414 it is not clear the 95%. Is it better then regression model or 95% is the NN accuracy?

R. We acknowledge this comment. We re-wrote this information as follows (L 651)

….. superior to the regression model with an accuracy significant level (95%).

Q 14. What is load in line 415?

R. We acknowledge this comment. We change load for electrical consumption

Q 15. In line 428 what is 1sec? Training or testing?

R. We acknowledge this comment. We re-wrote this line as follows (L 665-667):

The hidden layer of the network consisted of 140 neurons, testing was obtained in 1 second and prediction

Q16. In the item 6 it would be interesting a comparation between all algorithms and results presented in the article. It would improve the understanding the topic. It is important to evaluate all the presented results.

R. We acknowledge this comment. We added new information in section 6. In addition, information was added in section 2, where the characteristics of the NN presented are compared.

I hope such comments could help you the article improvements.

Round 2

Reviewer 1 Report

Although the authors have made some revisions to the review, the new version is not a major revision of the previous version as it should have been.

Although the current paper summarizes concepts and a great  amount of related work, it does not address them critically. What are the main questions that the authors are trying to answer with the review? What are the main findings from this review? Does the literature provide support to the claims that NN can model microclimate? If so, what would be the challenges for this particular application? Or is it only a matter of giving the new input and the model will be better than the others just by training it? What are the trade-offs?

The paper does not create connections among the different concepts or the different papers. I think that the tables are a great start but more critical review is needed for the paper to be published as a review.

This is my main concern with the paper.

On the other hand, some of the replies to my previous comments are not sufficient or reply only to my single example but the authors didn't revise the full paper.

For example,

NN "is very suitable for modeling dynamic systems in real time " does not explain why this model is better than others. Neural networks, depending on the number of layers and neurons, can also be considered as black box models. As other machine learning models, they can be subject to partial optimization, and can be fragile when the input varies from that of the training, specially in deep learning. See for example https://www.nature.com/articles/d41586-019-03013-5

The authors have stated that they have data, but is it enough?

Still many arguments are given without support.

Reviewer 2 Report

The authors have made significant modifications to the manuscript that also addressed most of my concerns. I think it can be considered for publication with some minor proof-reads and spell-checks. 

Reviewer 3 Report

Thank you very much to send this revised work. Several questions were answered, however some questions still persist and new questions arise due to the new texts.

Figure 1 did not really helps to understand neural networks - it is something to text books and does not contribute to better understand your work. the same about lines 95 to 100.

Between lines 99 and 101 it is presented a dataset (number of rows/cases)  and test distribution. There are several questions about this lines:

-1 why to make a distribution 25% x 25% - kfold could be better? Justify such choice.

- 2 there are 70032 cases - what is the quality of such dataset? such cases does really represent the studied phenomenon?

- 3 why such information is relevant to the topic of this work?

line 338 - .. greenhouse micrioclimate prediction is better?

in table - Dariouchv [86] - same parameter as input and output?

page 21 - blank? page 25 in landscape?

line 928 - not clear statement

lack of space between 9698 and 969

I'm really confused about some terms used in this work. it is not clear if your target is Agriculture 4.0, PA or SA. What is the difference between them and why do you need to use all of them? For example, there is a difference between PA and agriculture 4.0 (line 912). In line 772 it is clear that Agriculture 4.0 and SA are not the same thing - but it is not clear what is such difference. In line 932 it is clear the difference between SA and PA - "... broadens the concept of PA..." But is not clear the relationship between the 3 concepts (something important for a review) - and in the end if line 912 is still valid and how to relate Agriculture 4.0 and SA.

line 969 is a new item or an explanation?

I hope you can better describe such points. the work is valid and important to the community understand the trends and impacts of computing techniques in agriculture